# Hepatocyte growth factor derived from senescent cells attenuates cell competition-induced apical elimination of oncogenic cells

Nanase Igarashi[1,2,8], Kenichi Miyata[1,3,8], Tze Mun Loo[1,8], Masatomo Chiba[1], Aki Hanyu[1], Mika Nishio[1], Hiroko Kawasaki[1], Hao Zheng[4], Shinya Toyokuni [4], Shunsuke Kon[5], Keiji Moriyama [2], Yasuyuki Fujita[6] & Akiko Takahashi [1,3,7✉]

Cellular senescence and cell competition are important tumor suppression mechanisms that restrain cells with oncogenic mutations at the initial stage of cancer development. However, the link between cellular senescence and cell competition remains unclear. Senescent cells accumulated during the in vivo aging process contribute toward age-related cancers via the development of senescence-associated secretory phenotype (SASP). Here, we report that hepatocyte growth factor (HGF), a SASP factor, inhibits apical extrusion and promotes basal protrusion of Ras-mutated cells in the cell competition assay. Additionally, cellular senescence induced by a high-fat diet promotes the survival of cells with oncogenic mutations, whereas crizotinib, an inhibitor of HGF signaling, provokes the removal of mutated cells from mouse livers and intestines. Our study provides evidence that cellular senescence inhibits cell competition-mediated elimination of oncogenic cells through HGF signaling, suggesting that it may lead to cancer incidence during aging.

[1] Project for Cellular Senescence, Cancer Institute, Japanese Foundation for Cancer Research, Tokyo, Japan. [2] Department of Maxillofacial Orthognathics, Tokyo Medical and Dental University, Tokyo, Japan. [3] Cancer Cell Communication Project, NEXT-Ganken Program, Japanese Foundation for Cancer Research, Tokyo, Japan. [4] Department of Pathology and Biological Responses, Nagoya University Graduate School of Medicine, Nagoya, Japan. [5] Division of Development and Aging, Research Institute for Biomedical Sciences, Tokyo University of Science, Chiba, Japan. [6] Department of Molecular Oncology, Graduate School of Medicine, Kyoto University, Kyoto, Japan. [7] Advanced Research & Development Programs for Medical Innovation (PRIME), Japan Agency for Medical Research and Development (AMED), Tokyo, Japan. [8] These authors contributed equally: Nanase Igarashi, Kenichi Miyata, Tze Mun Loo. ✉email: akiko.takahashi@jfcr.or.jp

Somatic mutations in some oncogenes or tumor suppressor genes lead to cancer. At the earliest stage of cancer development, mutated damaged cells are selectively removed by several tumor-suppressive mechanisms, such as cell competition, cellular senescence, and various types of cell death pathways[1–3]. Although developmental studies on *Drosophila* were the first to report about cell competition, recent studies have shown that cells with oncogenic mutations compete with surrounding normal cells and are apically excluded from the epithelial layer in mammalian organs[4–6]. In contrast, cellular senescence is a phenomenon in which somatic cells display a finite replicative lifespan under in vitro culture conditions, which inhibits the proliferation of cancer-prone cells in precancerous lesions and benign tumors[2,7,8].

The functional decline of various tissues and organs due to aging leads to the development of multiple types of diseases. The status of cell fitness also changes during aging[9,10]. Skin homeostasis is maintained via epidermal stem cell competition, whereas epidermal stem cell aging leads to reduced cell competition, resulting in skin aging[11]. In addition, aging is a high-risk factor for cancer incidence. However, the effects of aging on cell competition as a tumor-suppressive mechanism remain unelucidated. Oncogenic mutations, including Ras signaling-related mutations, induce cell competition and cellular senescence. Therefore, they seem to occur simultaneously, inhibiting the proliferation of mutated cells as a barrier against tumor formation. Recent studies have exhibited the harmful side effects of cellular senescence that occurs during aging. Senescent cells accumulate throughout the body during the aging process[12–14] and secrete proinflammatory factors, including cytokine, chemokine, modifiers of the extracellular matrix, and growth factors. This phenotype, termed as senescence-associated secretory phenotype (SASP), promotes cancer malignancy at the initiation or progression stages and contributes toward the development of numerous other age-related pathologies[15,16].

Here, we explored the function of a SASP factor, hepatocyte growth factor (HGF), in cell competition and discovered that HGF inhibits apical extrusion and promotes basal protrusion of Ras-mutated cells using in vitro and in vivo cell competition models. Furthermore, these data demonstrate that cellular senescence inhibits cell competition-induced elimination of oncogenic cells through HGF signaling. This mechanism may increase the risk of cancer development in the aging population.

## Results

**Secreted factors from senescent fibroblasts inhibit the apical extrusion of Ras$^{V12}$ epithelial cells.** To examine the effect of SASP factors on cell competition, a conditioned medium (CM) was prepared from human diploid fibroblasts, IMR-90 cells, which were subjected to senescence via H-Ras$^{V12}$ expression. The induction of cellular senescence was confirmed via an observed decrease in LMNB1, an increase of senescence markers p16$^{INK4a}$ and SASP factors (IL-6, IL-8, and CXCL10), senescence-associated β-galactosidase (SA-β-Gal) staining with loss of Ki-67, a feature of senescent cell cycle arrest, and the formation of DNA damage foci (Fig. 1a–d). In line with the illustrated time course in Fig. 1e, parent Madin–Darby canine kidney cells (MDCK) and green fluorescent protein (GFP)-tagged H-Ras$^{V12}$-expressing MDCK cells (MDCK-pTR GFP-Ras$^{V12}$) were treated with CM derived from control or Ras$^{V12}$-induced senescent IMR-90 cells for 3 days, followed by cell competition assay on the collagen gel-coated plate to mimic physiological tissues[17,18]. After treatment with CM derived from proliferating IMR-90 cells, similar to nontreated cells, MDCK-pTR GFP-Ras$^{V12}$ cells were extruded by surrounding MDCK cells. However, CM derived from Ras$^{V12}$-induced senescent IMR-90 cells promoted the

nonextruded population (Fig. 1f), similar to CM derived from IMR-90 or TIG-3 cells, which were subjected to senescence via serial passages (Supplementary Figs. 1 and 2a–c), oncogene overexpression (H-Ras$^{V12}$) (Supplementary Fig. 2d–f), or X-ray irradiation (Supplementary Figs. 2g–i and 3a–c). Similarly, CM derived from senescent IMR-90 or TIG-3 cells promoted the nonextruded population, although CM derived from non-senescent cells did not affect the status of MDCK-pTR GFP-Ras$^{V12}$ cells (Supplementary Figs. 1e, 2j, and 3d). These data suggest that senescent CM containing SASP factors inhibits the apical extrusion of Ras$^{V12}$ MDCK cells. However, it is unknown whether CM affects surrounding cells (MDCK cells) or oncogenic cells (MDCK-pTR GFP-Ras$^{V12}$ cells) in this model. To elucidate this point, a cell competition assay was performed using senescent CM-treated MDCK cells (parent) with nontreated MDCK-pTR GFP-Ras$^{V12}$ cells and vice versa (Fig. 1g). We found that treatment with senescent CM did not affect the apical extrusion of parent cells. In contrast, senescent CM remarkably suppressed the frequency of apical extrusion of Ras$^{V12}$ cells from the epithelial monolayer (Fig. 1h), indicating that CM derived from senescent cells decreased the cell competition capacity of Ras$^{V12}$ cells.

**HGF inhibits the apical extrusion of Ras$^{V12}$ epithelial cells.** To identify the factor responsible for the inhibition of cell competition, we conducted a cytokine array using CM derived from proliferating and Ras$^{V12}$- or X-ray-induced senescent IMR-90 cells (Fig. 1 and Supplementary Fig. 3a–d). Based on the cytokine array analysis, 10 candidates were identified, which were commonly upregulated in both CM derived from senescent cells compared with that from proliferating cells (Fig. 2a, b and Supplementary Fig. 3e). By applying each candidate peptide or dichloroacetate, which suppresses the apical extrusion of Ras$^{V12}$-transformed cells[18], it was discovered that HGF significantly inhibited the apical extrusion of MDCK-pTR GFP-Ras$^{V12}$ cells (Fig. 2c and Supplementary Fig. 3f). To confirm the effect of HGF on cell competition, the concentration of HGF in X-ray-induced senescent CM was quantified (Fig. 2d) and the equivalent concentration of HGF (20 ng/ml) was enough to inhibit the apical extrusion of Ras$^{V12}$ MDCK cells (Fig. 2e). Surprisingly, HGF treatment upregulated the rate of cell protrusion into the basal layer (Fig. 2c, e and Supplementary Fig. 3f). Conversely, CM derived from HGF-depleted senescent cells using small interfering RNAs (siRNAs) increased the percentage of apically extruded cells harboring H-Ras$^{V12}$ mutation (Fig. 2f, g). HGF secretion significantly increased in all senescent IMR-90 and TIG-3 cells, regardless of the route (Fig. 2d and Supplementary Figs. 1f and 2k), inhibiting the apical extrusion of MDCK-pTR GFP-Ras$^{V12}$ cells (Fig. 1f and Supplementary Figs. 1e, 2j, and 3d). Altogether, these data strongly indicate that HGF in SASP factors inhibits apical extrusion and promotes basal protrusion of Ras$^{V12}$ cells.

**HGF induces epithelial–mesenchymal transition (EMT) of Ras$^{V12}$ epithelial cells.** Next, to investigate the molecular mechanisms involved in the inhibition of cell competition by SASP factors, we assessed the change in the gene expression profiles of CM-treated MDCK-pTR GFP-Ras$^{V12}$ cells using a microarray gene chip. In Ras$^{V12}$ MDCK cells treated with Ras$^{V12}$- or X-ray induced senescent CM, 944 or 1327 genes, respectively, were upregulated by more than twofold compared to control CM-treated cells (Fig. 3a). Among the 247 genes that were commonly upregulated in both the types of senescent CM-treated Ras$^{V12}$ MDCK cells, EMT-associated gene profile was changed globally using Gene Ontology (GO) analysis (Fig. 3a). We observed that Ras$^{V12}$ cells treated with senescent CM exhibited typical changes related to EMT-associated morphology (Fig. 3b). In addition, the

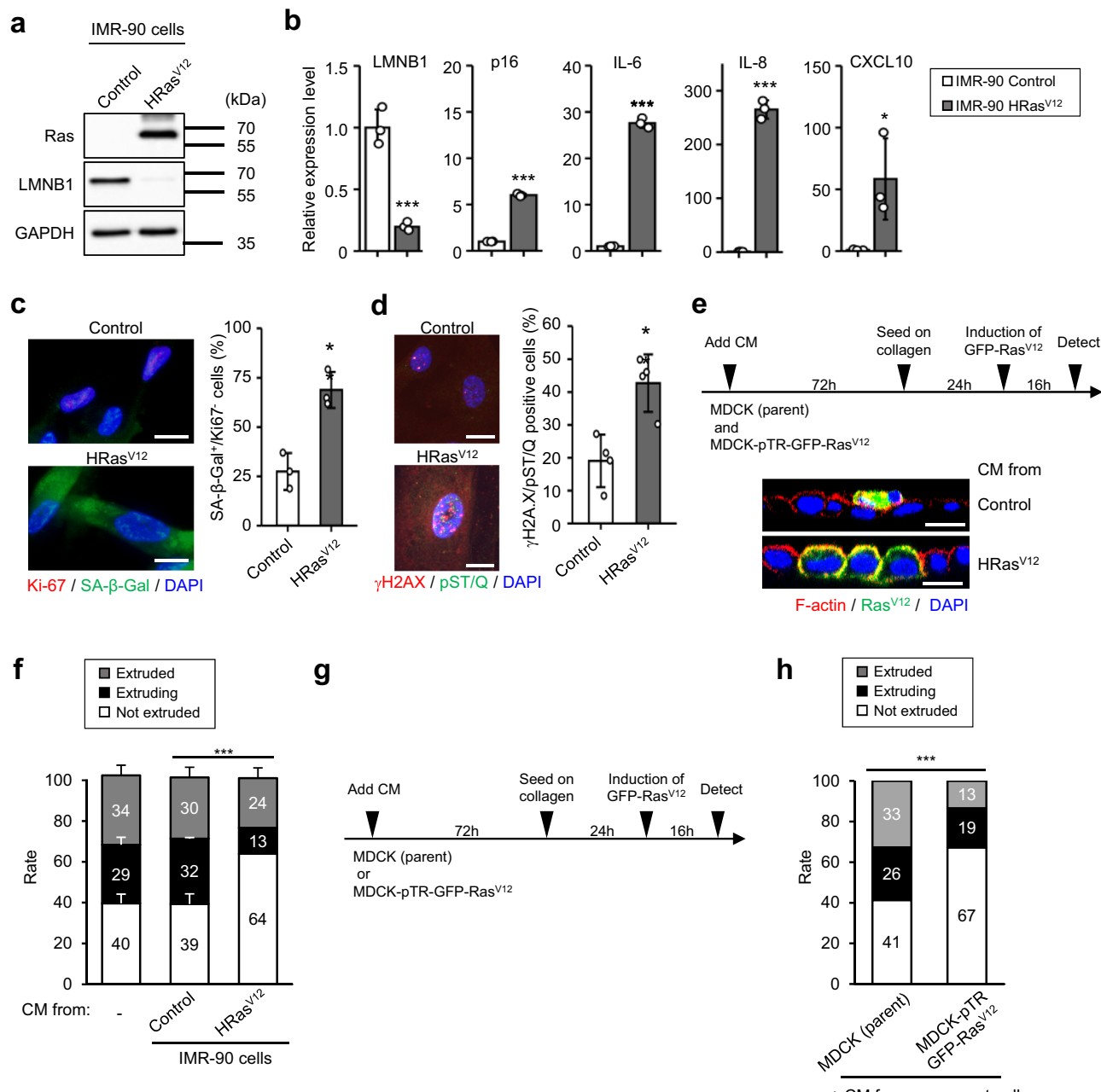

**Fig. 1 The secretome from senescent cells suppresses apical extrusion of oncogenic Ras$^{V12}$-expressing MDCK cells from the surrounding normal epithelium. a–c** Presenescent IMR-90 cells were rendered senescent by ectopic expression of oncogenic *ras* (H-Ras$^{V12}$). These cells were then subjected to western blotting using antibodies shown toward the left (**a**), RT-qPCR analysis of p16$^{INK4a}$ and SASP factor gene expression (**b**) (LMNB1, $P < 0.001$; p16, $P < 0.001$; IL-6, $P < 0.001$; IL-8, $P < 0.001$; CXCL10, $P = 0.039$), or immunofluorescence staining for markers of senescent cell cycle arrest (Ki-67 (red), SA-β-gal (green) and DAPI (blue), $P = 0.007$) (**c**) and DNA damage (γ-H2AX (red), phosphor-Ser/Thr ATM/ATR (pST/Q) substrate (green) and DAPI (blue), $P = 0.005$) (**d**). The blotting experiments have performed at least two times (**a**). Representative data from three independent experiments are shown (**b–d**). The histograms indicate the percentage of Ki-67-negative and SA-β-gal-positive cells (**c**) and nuclei containing more than three foci positive for γ-H2AX and pST/Q staining. At least 100 cells were scored per group (**c, d**). Scale bar, 10 µm. **e–h** MDCK (parent) and MDCK-pTR GFP-Ras$^{V12}$ cells were separately treated with CM derived from proliferating, or oncogene-induced senescent IMR-90 cells ($P < 0.001$) (**e, f**) or CM derived from oncogene-induced senescent IMR-90 cells ($P < 0.001$) (**g, h**) for 3 days. The treated MDCK (parent) and MDCK-pTR GFP-Ras$^{V12}$ cells were mixed at a ratio of 50:1 and cultured on type-I collagen gels. These MDCK cells were stained with phalloidin (F-actin, red) and DAPI (blue) after 16 h of incubation with tetracycline to induce Ras$^{V12}$ in MDCK-pTR GFP-Ras$^{V12}$ cells. Representative confocal images of *xz* sections of Ras$^{V12}$-expressing MDCK cells (green) in a monolayer or normal MDCK cells (**e**). Scale bar, 10 µm. Quantification of the apically extruded, extruding, or not extruded MDCK-pTR GFP-Ras$^{V12}$ cells from a monolayer of MDCK normal cells (**f, h**). Representative data from three independent experiments are shown. Error bars indicate the mean ± standard deviation of three independent measurements for all graphs. *$P < 0.05$, **$P < 0.01$, or ***$P < 0.001$ by the unpaired two-sided *t* test (**b–d**) or the chi-square test (**f, h**).

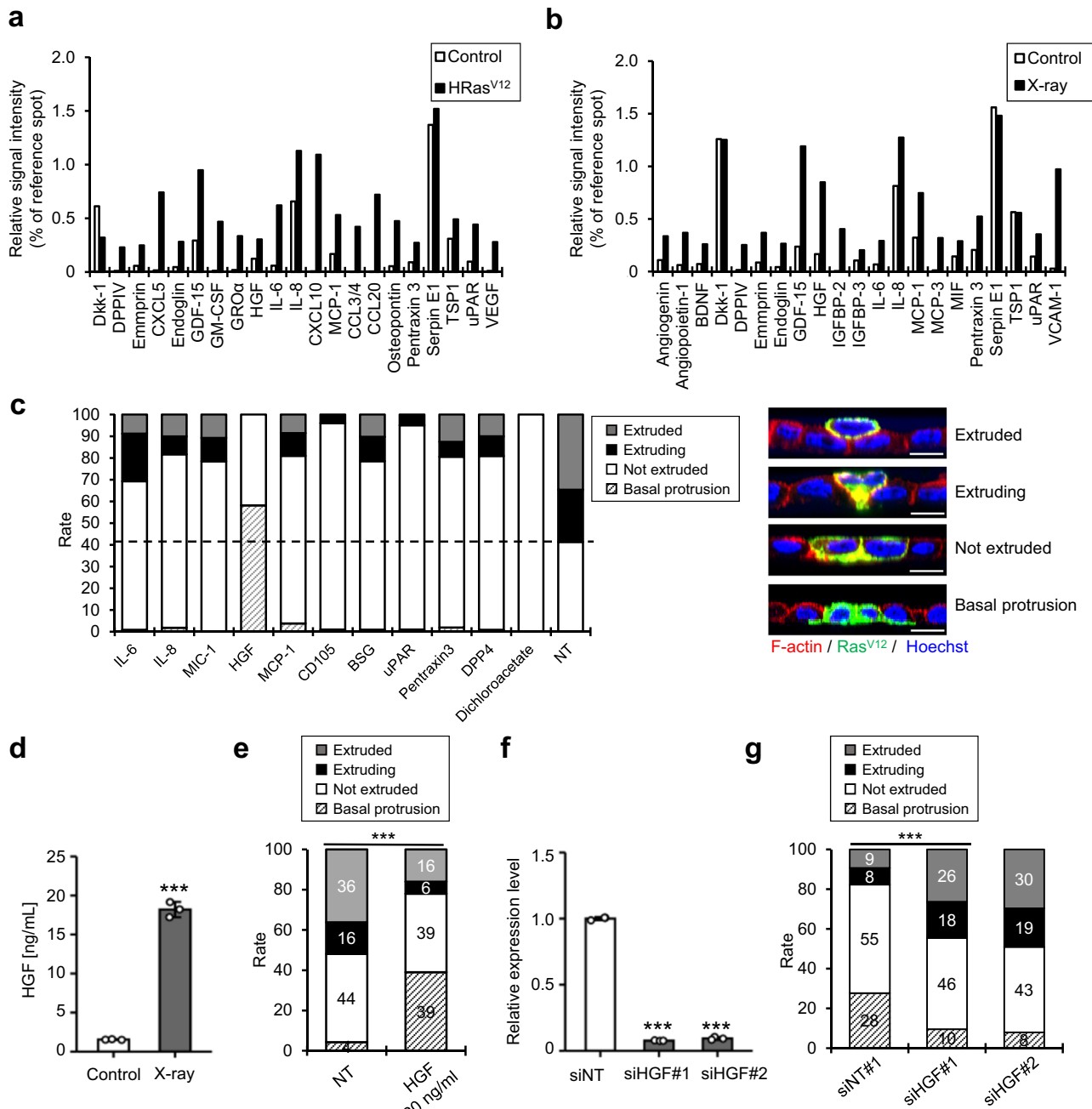

**Fig. 2 HGF suppresses apical elimination of Ras^V12 cells. a**, **b** Cytokine array analysis of CM derived from oncogene- (**a**) and X-ray-induced (**b**) senescent IMR-90 cells (**a**, **b**, $n = 1$). Human cytokine array analysis of multiple cytokines secreted by MDCK-pTR GFP-Ras^V12 cells after treatment with CM derived from senescent IMR-90 cells. Protein levels are shown as the relative spot intensity (average signal intensity of each spot divided by the control array). **c** MDCK and MDCK-pTR GFP-Ras^V12 cells were treated with 250 ng/ml of each recombinant protein or 50 μM dichloroacetate, which suppresses the apical extrusion of Ras^V12-transformed cells[17], for 3 days and mixed at a ratio of 50:1 and cultured on type-I collagen gels. After 16 h of incubation with tetracycline to induce Ras^V12 in MDCK-pTR GFP-Ras^V12 cells, quantification of the apically extruded, extruding, not extruded or basally protruded MDCK-pTR GFP-Ras^V12 cells from a monolayer of MDCK normal cells was performed. NT nontreatment. Scale bar, 10 μm. **d** Analysis of HGF production in X-ray-induced senescent IMR-90 cells using ELISA. Ten days after senescence induction by 10-Gy irradiation, CM was harvested and subjected to ELISA for HGF production analysis. $P < 0.001$. **e** MDCK-pTR GFP-Ras^V12 cells were treated with 20 ng/ml (control) or 20 ng/ml HGF for 3 days before performing cell competition assay. $P < 0.001$. **f**, **g** Senescent IMR-90 cells induced by oncogenic *ras* expression were transfected with validated siRNA oligos against HGF twice at 2-day intervals. These cells were then subjected to RT-qPCR for analyzing the expression levels of HGF (**f**). The relative expression level was normalized by siNT cells (siHGF#1, $P < 0.001$; siHGF#2, $P < 0.001$). MDCK and MDCK-pTR GFP-Ras^V12 cells were treated with CM derived from si control or siHGF-treated oncogene-induced senescent IMR-90 cells for 3 days, mixed at a ratio of 50:1, and cultured on type-I collagen gels. After 16 h of incubation with tetracycline to induce Ras^V12 in MDCK-pTR GFP-Ras^V12 cells, quantification of the apically extruded, extruding, or not extruded MDCK-pTR GFP-Ras^V12 cells from a monolayer of MDCK normal cells was performed (**g**) ($P < 0.001$). Representative data from three independent experiments are shown (**d**, **e**, **g**). Error bars indicate the mean ± standard deviation of three independent measurements for all graphs. **$P < 0.01$, or ***$P < 0.001$ by unpaired two-sided *t* test (**d**), chi-square test (**e**, **g**), or one-way ANOVA followed by Dunnett's multiple comparisons posthoc test (**f**).

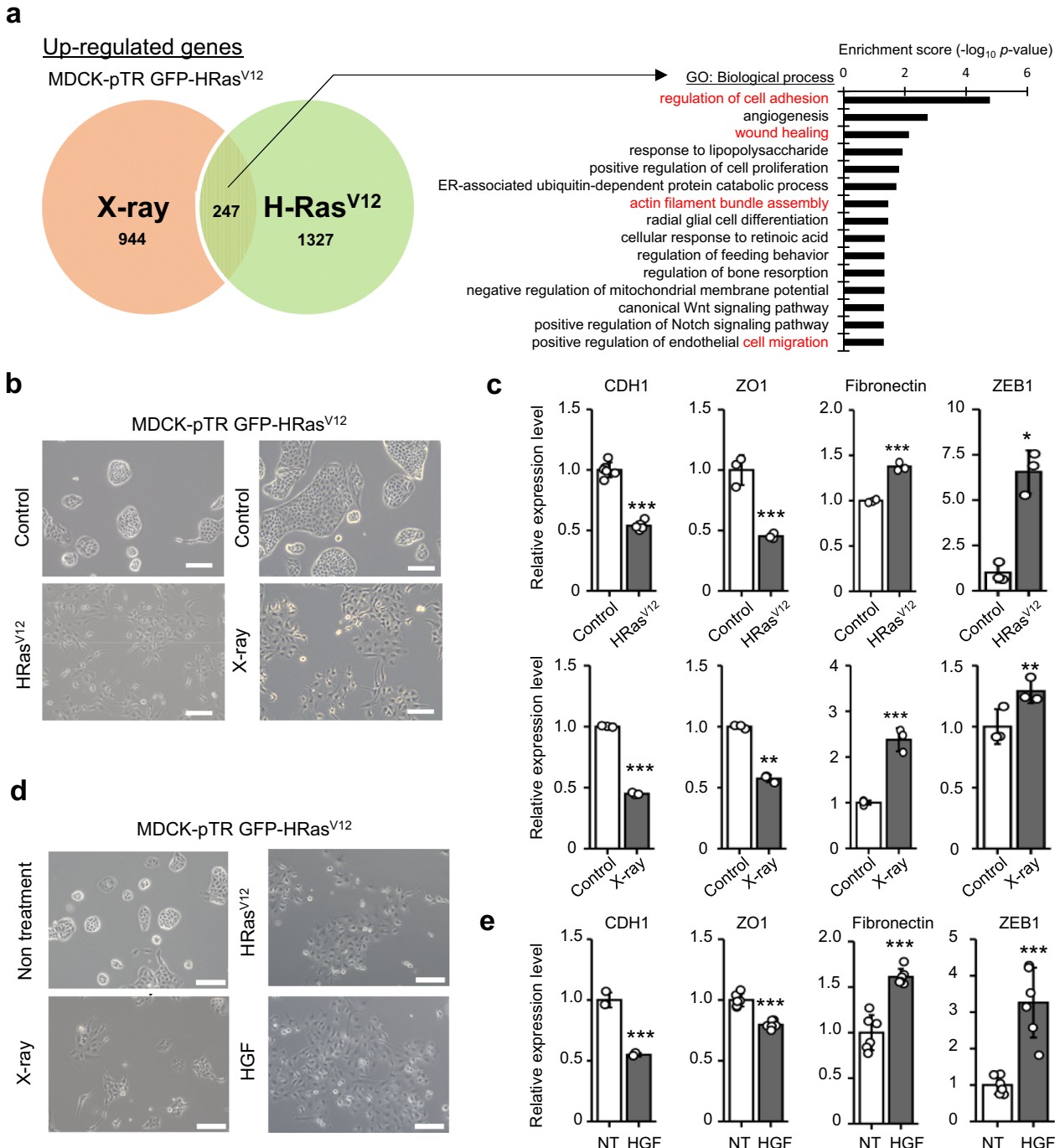

**Fig. 3 HGF derived from senescent cells induces EMT of Ras$^{V12}$ MDCK cells. a** Venn diagram indicates upregulated genes that presented at least a twofold significant change in expression in MDCK-pTR GFP-Ras$^{V12}$ cells treated with CM of X-ray- (red) and oncogene-induced senescent IMR-90 cells (green) than in control cells. Out of 2518 genes detected in two samples, 247 were commonly upregulated in both. An enrichment test using GO analysis was performed with those commonly upregulated genes in MDCK-pTR GFP-Ras$^{V12}$ cells. The 15 most enriched terms within the biological process of GO are described in the bar graph. **b** The appearance of MDCK-pTR GFP-Ras$^{V12}$ cells treated with CM derived from oncogene- or X-ray-induced senescent IMR-90 cells for 24 h. Scale bar, 100 μm. **c** RT-qPCR analysis of EMT markers in treated MDCK-pTR GFP-Ras$^{V12}$ cells (H-Ras:$^{V12}$ CDH1, $P < 0.001$; ZO1, $P = 0.0016$; fibronectin, $P < 0.001$; ZEB1, $P = 0.0018$), (X-ray: CDH1, $P < 0.001$; ZO1, $P < 0.001$; fibronectin, $P < 0.001$; ZEB1, $P = 0.044$). **d** The appearance of nontreated MDCK-pTR GFP-Ras$^{V12}$ cells, MDCK-pTR GFP-Ras$^{V12}$ cells treated with CM derived from oncogene-induced senescent IMR-90 cells, X-ray-induced senescent IMR-90 cells, or cells treated with 20 ng/ml HGF for 24 h. Scale bar; 100 μm. **e** RT-qPCR analysis of EMT markers in MDCK-pTR GFP-Ras$^{V12}$ cells treated with 20 ng/mL HGF (CDH1, $P < 0.001$; ZO1, $P < 0.001$; fibronectin, $P < 0.001$; ZEB1, $P < 0.001$). CDH1 and ZO1 are shown as epithelial markers (**c**, **e**). Fibronectin and ZEB1 are shown as mesenchymal markers (**c**, **e**). Representative data from three independent experiments are shown. Error bars indicate the mean ± standard deviation of three independent measurements for all graphs. *$P < 0.05$, **$P < 0.01$, or ***$P < 0.001$ by the unpaired two-sided $t$ test (**c**, **e**).

expression levels of epithelial markers (such as E-cadherin (CDH1) and zonula occludens protein-1 (ZO1)) decreased and that of mesenchymal markers (fibronectin and Zinc finger E-box-binding homeobox 1 (ZEB1)) increased in Ras$^{V12}$ cells treated with senescent CM (Fig. 3c)[19,20]. By adding 10 candidate peptides identified through cytokine array analysis to MDCK-pTR GFP-Ras$^{V12}$ cells (Fig. 2c), only HGF treatment resulted in the same phenotype as that produced by senescent CM treatment (Fig. 3d and Supplementary Fig. 4). Moreover, as previously described, HGF promoted EMT-like gene expression in MDCK cells (Fig. 3e)[21]. To further clarify the involvement of EMT in cell competition, we examined the effect of TGF-β1, a well-established inducer of EMT[22,23]. We found that TGF-β1 also inhibited apical extrusion and promoted basal protrusion of MDCK-pTR GFP-Ras$^{V12}$ cells (Supplementary Fig. 5). These data suggest that HGF leads to EMT-like phenotypic changes in Ras$^{V12}$ cells and suppresses cell competition.

**Crizotinib treatment improves the efficiency of cell competition in HFD-induced senescent tissues.** HGF is a ligand of HGF receptor (c-Met). It activates downstream signaling pathways and promotes carcinogenesis and cancer progression[24,25]. We used crizotinib, a small molecule inhibitor of HGF/c-Met signaling, in an in vitro cell competition assay. Crizotinib treatment was found to significantly inhibits c-Met signaling induced by HGF and recover the rate of apically extruded Ras$^{V12}$ cells (Supplementary Fig. 6a, b). In addition, crizotinib treatment prevented the senescent CM-induced inhibition of cell competition (Supplementary Fig. 6c), which indicates that crizotinib can inhibit the negative impact of SASP factors on cell competition.

To confirm the effect of cellular senescence on cell competition in vivo, we used a cell competition mouse model fed a high-fat diet (HFD)[26,27]. In the obese mice model, HFD induces senescence in mice hepatic stellate cells (HSCs) via deoxycholic acid (DCA) produced by the gut microbiota[28–31]. First, we checked the expression levels of HGF in human and mouse HSCs. Cellular senescence was induced by treatment with DCA and lipoteichoic acid (LTA), a major constituent of the cell wall of gram-positive bacteria, as previously described[29,30]. After DCA/LTA treatment, senescent HSCs exhibited a significant increase in HGF expression (Supplementary Fig. 7a, b). It was previously reported that obesity caused by HFD increases SASP factor gene expression in HSCs, triggering the promotion of hepatocellular carcinoma development in obese mice after exposure to a chemical carcinogen[28–31]. Indeed, HGF mRNA and protein expression levels significantly increased in senescent HSCs in the livers of obese mice than in those in the livers of lean mice fed a normal diet (ND) (Supplementary Fig. 8a–c). In addition, we observed an appearance of some senescence markers, such as increased CDK inhibitor (p21$^{Cip1/Waf1}$, p16$^{INK4a}$) and SASP factor (CXCL10) expression, DNA damage accumulation (53BP1), and SA-β-Gal activity (Supplementary Fig. 8a–c).

Next, hydrodynamic tail vein injection (HTVi) was performed to induce GFP-N-Ras$^{V12}$-internal ribosomal entry sites (IRES)-luciferase expression in the livers of ND- and HFD-fed mice (Fig. 4a, b). According to a previous report, HTVi induces a mosaic gene expression by introducing plasmids in approximately 30% of hepatocytes (Fig. 4c)[26]. In vivo imaging analysis performed to monitor luciferase luminescence demonstrated that the number of Ras$^{V12}$-IRES-luciferase-expressing hepatocytes decreased on day 6 compared to day 1 after injection (Fig. 4d). In contrast, Ras$^{V12}$-expressing hepatocytes remained in the liver of obese mice, and the frequency of eliminated cells did not change for 5 days (Fig. 4c, d); this finding was similar to that of a previous report that used transgenic mouse models for evaluating cell competition[27]. In addition, Ras$^{V12}$-expressing hepatocytes in HFD-fed mice exhibited

profiles typical of EMT cells, such as decreased expression of epithelial marker protein (E-cadherin) and increased expression of mesenchymal marker protein (vimentin) (Supplementary Fig. 9a, b). The administration of crizotinib, a c-Met tyrosine kinase inhibitor, reduced the remaining population of Ras$^{V12}$-expressing hepatocytes in obese mice but not in control mice (Fig. 4c, d and Supplementary Fig. 9c).

A previously established cell competition mouse model was used to confirm this phenomenon in another cell competition model. In these models, *LSL-Ras$^{V12}$-IRES-enhanced GFP* (eGFP) and villin-Cre-ERT2 mice and its organoid exhibit Cre-dependent induction of Ras$^{V12}$ expression, which is evaluated through the simultaneous expression of eGFP[18]. Using this system, we examined the effect of cellular senescence on cell competition ex vivo and in vivo. Intestinal organoids established from cell competition mouse model were cocultured with nonsenescent or senescent IMR-90 cells. Coculturing organoids with senescent fibroblasts significantly inhibited apical extrusion and promoted basal extrusion of Ras$^{V12}$-transformed epithelial cells compared with organoids with nonsenescent fibroblasts (Supplementary Fig. 10). Finally, we examined the inhibitory effect of HGF from senescent cells on cell competition in vivo. After feeding ND or HFD for 3 months, the administration of low-dose tamoxifen causes mosaic expression of Ras$^{V12}$ in intestinal epithelia (Fig. 5a and Supplementary Fig. 11a). Although Ras$^{V12}$-transformed cells were apically extruded into the apical lumen in mice fed a ND, HFD was found to attenuate cell competition[27]. However, crizotinib suppressed the inhibitory effect of HFD and improved the elimination efficiency of cells with oncogenic mutations in the small intestine (Supplementary Fig. 11b–d). To further explore the involvement of senescent cells in cell competition, we examined the effect of an established senolytic drug, ARV825, which selectively eliminates senescent cells from obese mice[31]. ARV825 treatment notably elevated the frequency of apically extruded Ras$^{V12}$-expressing cells in HFD-fed obese mice (Fig. 5b, c). These data demonstrate that HFD-induced cellular senescence inhibits cell competition-mediated elimination of Ras$^{V12}$-transformed cells from the epithelial layers.

## Discussion

Previous studies demonstrated that HFD treatment-induced chronic inflammation inhibits apical extrusion of Ras-mutated cells in the small intestine and pancreas[27]. In addition, chronic inflammation promotes senescence of epithelial cells, fibroblasts, and immune cells in various tissues[2,28–31]. Therefore, we speculated that soluble factors secreted by senescent cells would influence cell competition. However, the molecular mechanism underlying this phenomenon has not been elucidated. This study reveals that HGF derived from senescent fibroblasts inhibits the cell competition activity of transformed cells in the epithelial layer.

Cell competition efficiently removes cancer-prone "loser" cells from normal epithelial layers to maintain epithelial homeostasis. Cytoskeletal rearrangements of normal epithelial "winner" cells are required for the extrusion of Ras$^{V12}$-expressing MDCK cells or *scrib*-expressing *Drosophila* cells[17,32]. The appropriate balance of mechanical stress between loser and winner cells is essential for competition-mediated apical extrusion of loser cells, implying that junctional changes are important for regulating cell competition[33–35]. Our data indicate that HGF secreted by senescent cells induces EMT-like morphological change and promotes basal protrusion of Ras$^{V12}$-mutated cells. The deficiency of cell polarity and/or junctional remodeling caused by EMT in Ras$^{V12}$-mutated cells might lead to cell competition failure. Consequently, Ras$^{V12}$-mutated cells proliferate in the epithelial layers and gain the potential to invade basal layers.

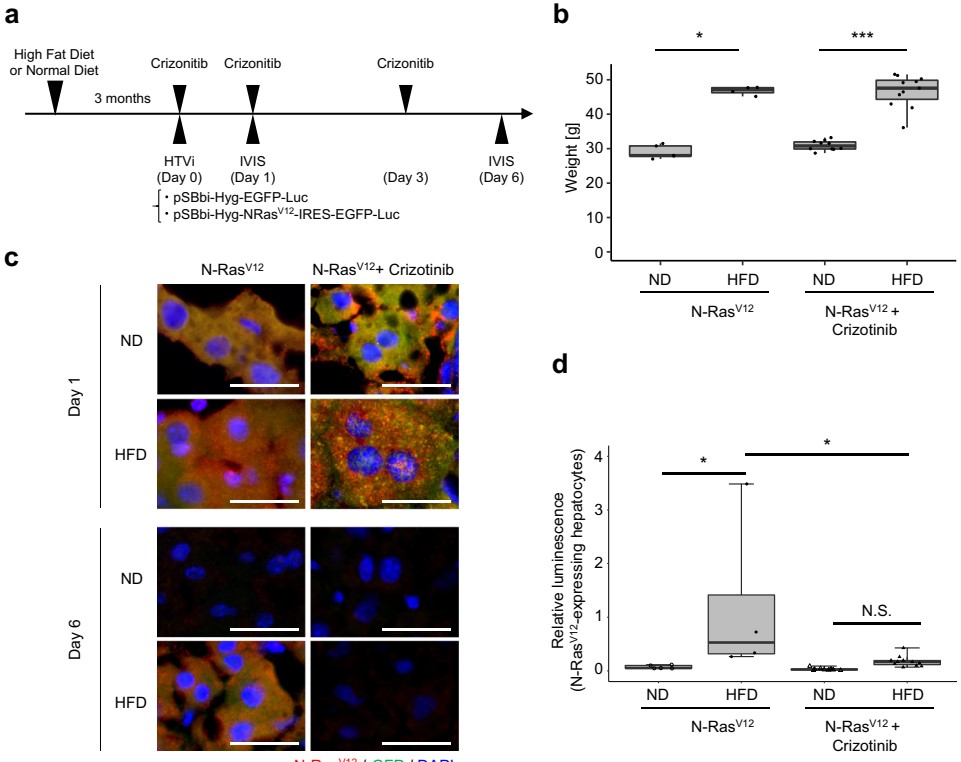

**Fig. 4 Crizotinib improves the efficiency of cell competition in HFD-fed mouse liver. a** Timeline of the experimental procedure. C57BL/6 mice were fed ND or HFD for 3 months and subjected to HTVi with empty control or plasmid-encoding GFP-N-Ras$^{V12}$-IRES-luciferase (N-Ras$^{V12}$, $n = 5$ (ND), $n = 4$ (HFD); N-Ras$^{V12}$ + Crizotinib, $n = 10$ (ND), $n = 11$ (HFD)). Some HFD-fed mice were orally administered 50 mg/kg crizotinib thrice. The mice were euthanized and subjected to in vivo bioluminescent imaging to confirm the GFP-N-Ras$^{V12}$-IRES-luciferase expression at days 1 and 6. **b** The average body weight of each group on day 6 in (**a**) (N-Ras$^{V12}$, $n = 5$ (ND), $n = 4$ (HFD); N-Ras$^{V12}$ + Crizotinib, $n = 10$ (ND), $n = 11$ (HFD)). N-RasV12: HFD vs ND, $P = 0.016$, N-RasV12 + Crizotinib: HFD vs ND, $P < 0.001$. **c** Immunofluorescence analysis of liver sections stained using N-Ras (red) and cell nuclei stained using DAPI (blue). Scale bar, 20 μm. **d** The values of relative luminescence in each group (see "Methods"). The relative luminescence was normalized by the mice subjected to HTVi with empty control and ND-fed mice. HFD (N-Ras$^{V12}$) vs (ND) N-Ras$^{V12}$, $P = 0.017$; HFD (N-Ras$^{V12}$) vs HFD (N-Ras$^{V12}$ + Crizotinib), $P = 0.013$. *$P < 0.05$, ***$P < 0.001$, or not significant (N.S.) by Wilcoxon rank-sum test (**b**), or one-way ANOVA, followed by Tukey's multiple comparisons posthoc test (**d**).

Further investigation of the precise molecular mechanism underlying EMT of Ras$^{V12}$-mutated cells would provide us with a potential target for the regulation of cell competition.

The expression of oncogenes, such as Ras$^{V12}$, causes normal cells to senesce, acting as an important barrier against transformation[2,7]. HTVi-induced N-Ras$^{V12}$ expression in the mouse liver was reported to induce oncogene-induced senescence in hepatocytes[36,37]. In this model, senescent hepatocytes secrete various chemokines and cytokines, triggering senescence surveillance by monocytes, macrophages, and CD4 + T cells 12 days after injection and enabling immune-mediated clearance of senescent cells 60 days after injection[37]. The loss of Ras$^{V12}$-expressing hepatocytes in ND-fed mice was confirmed 6 days after injection, as previously described[26]. These data suggested that cell competition-mediated loss of oncogenic cells occurs before immune cell-mediated senescence clearance. On the contrary, significantly more Ras$^{V12}$-expressing cells were detected in the liver of HFD-fed mice than in the liver of ND-fed mice 7 days after injection, which was inconsistent with the finding of a previous report[27]. In HFD-fed mice, increased levels of DCA, a metabolite derived from the altered gut microbiota of obese individuals, induced cellular senescence and inflammatory SASP factor production in HSCs[28–31]. Our data demonstrated that crizotinib treatment significantly enhanced the elimination efficiency of Ras$^{V12}$-mutated cells in the liver of the HTVi model and the small intestine of the cell competition mouse model. Therefore, HGF secreted from senescent stromal fibroblasts suppressed, at least

partly, the apical extrusion of Ras$^{V12}$-mutated cells. These data suggested that cell competition leads to the apoptosis of transformed cells as a short-term response. Then, cellular senescence inhibits cell proliferation and promotes immune-mediated senescence surveillance as a mid-term response to cancer prevention.

HGF/c-Met signaling is involved in multiple cellular processes, including embryonic development, angiogenesis, and tumor progression in vivo[25,38]. Furthermore, many reports have demonstrated that the aberrant activation of HGF/c-Met signaling via somatic mutation, amplification, and SNPs promotes transformation, metastasis, stemness, and treatment resistance of various solid tumors[39]. Accordingly, HGF/c-Met signaling has been a prominent molecular target for cancer therapy. Crizotinib was clinically used for non-small cell lung cancer treatment, and numerous clinical trials are underway. Our study identified a pivotal function of crizotinib in reinforcing cell competition activity in the senescent environment, which significantly facilitated the frequency of apical extrusion of Ras$^{V12}$-mutated cells, thereby providing a potential therapeutic strategy for preventing the accumulation of transformed cells in the early stage of cancer development.

Since most premalignant mutant cells were eliminated via cell competition in a mouse model of esophageal carcinogenesis[40], cell competition could play an important role in tumor suppression during early carcinogenesis in mammals. In addition, the incidence of esophageal or other cancers dramatically increases

**Fig. 5 ARV825 improves apical extrusion of the Ras$^{V12}$-transformed cells in the small intestine of the HFD-fed cell competition model in vivo.**
**a** Timeline of the experimental procedure. Cell competition model mice, *LSL-Ras$^{V12}$-IRES-eGFP* and villin-Cre-*ERT* mice, in whom Ras$^{V12}$ expression is induced by tamoxifen treatment (TAM) in a Cre-dependent fashion and traced using simultaneous expression of eGFP[18, 27], were fed HFD for 3 months. The black arrow indicates an intraperitoneal treatment with 5 mg/kg ARV825. **b, c** Immunofluorescence images (**b**) and quantification analysis (**c**) of Ras$^{V12}$ cells in the epithelium of the small intestine. Phalloidin (F-actin, red) and Ras$^{V12}$ signals (green) were detected in the small intestine, and DNA was stained using DAPI (blue) (N-Ras$^{V12}$, $n = 3$ (ND); N-Ras$^{V12}$ + ARV825, $n = 3$ (ND); N-Ras$^{V12}$, $n = 3$ (HFD); N-Ras$^{V12}$ + ARV825, $n = 3$ (HFD)). Scale bars, 50 μm (HFD), 100 μm (ND). *$P < 0.05$, or not significant (N.S.) by the unpaired two-sided $t$ test.

with age, which indicates that aging is one of the critical risk factors for cancer development. Here, we show that treatment with the senolytic drug ARV825 improved the efficiency of apical extrusion of Ras$^{V12}$-mutated cells. Our discoveries highlight the crosstalk between two important tumor-suppressive mechanisms, cellular senescence, and cell competition, through SASP (Fig. 6). Aging causes the accumulation of senescent cells, and secreted SASP factors promote chronic inflammation in aged tissues; this leads to the attenuation of cell competition-induced apical removal of oncogenic cells, which may contribute to age-related carcinogenesis. Therefore, selective pharmacological targeting of the HGF/c-Met pathway or senescent cells may be a useful strategy for cancer prevention in aging individuals.

## Methods

**Cell culture.** TIG-3 cells[41,42] and IMR-90 cells[43] were obtained from the Japanese Cancer Research Resources Bank and American Type Culture Collection. TIG-3, IMR-90, and IMR-90/ER: H-Ras$^{V12}$ cells[44] were cultured in Dulbecco's Modified Eagle Medium (DMEM) (Nacalai Tesque) supplemented with 10% fetal bovine serum (FBS) and penicillin/streptomycin (Sigma-Aldrich) at physiological oxygen conditions (92% N$_2$, 5% CO$_2$, and 3% O$_2$) at 37 °C. Early passage TIG-3 cells (<40 population doublings) were used as proliferating (control) cells, and late passage TIG-3 cells (>70 population doublings) that stopped proliferating were used as replicative senescent cells. For X-ray-induced senescence, IMR-90 cells were exposed to 10-Gy irradiation using a CP-160 X-ray machine (Faxitron X-ray Corporation). After X-ray irradiation, IMR-90 cells were plated at a density of 2500 cells cm$^{-2}$. These cells were not passaged for 10 days after X-ray irradiation. Ras-induced senescence by 4-hydroxytamoxifen in IMR-90/ER: H-Ras$^{V12}$ cells was conducted as previously described[44]. CM was prepared by incubating proliferating (control) or senescent cells for 3 days in DMEM supplemented with 10% FBS. Proliferating (control) cells were plated at subconfluent density 1 day before starting CM preparation. CM was filtered through a 0.45-μm filter and then used for cell culture. MDCK (parent) and MDCK-pTR GFP-Ras$^{V12}$ cells were cultured according to the method described previously[17,18]. For human and mouse HSCs, senescence induction and cell culture were performed as described previously[29,30]. All cell lines used were negative for mycoplasma.

**Reagent and recombinant proteins.** The following reagents and recombinant proteins were used for cell competition assay: sodium dichloroacetate (Sigma-Aldrich, #347795); recombinant human IL-6 (PEPRO TECH, #200-06), IL-8 (PEPRO TECH, #200-08M), GDF-15/IMC-1 (PEPRO TECH, #120-28C), MCP-1 (CCL2, PEPRO TECH, #300-04), HGF (PEPRO TECH, #100-39), TGF-β1 (PEPRO TECH, #100-21), CD105 protein (Abcam, #ab54338), BSG protein (P01, Abnova, #H00000682-P01), DPP4 protein (P01, Abnova, #H00001803-P01), pentraxin 3/TSG-14 (R&D Systems, #1826-TS), and uPAR protein (84–95) (scrambled peptide, Anygen, #C180084).

**Cell competition assay.** Cell competition assay was performed as previously described[17,18] with minor modifications. In brief, after MDCK (parent) and MDCK-pTR GFP-Ras$^{V12}$ cells were treated with CM derived from proliferating (control) or senescent cells for 3 days (the details of which are provided in the "Cell culture" section), MDCK-pTR GFP-Ras$^{V12}$ cells were combined with normal MDCK (parent) cells at a ratio of 1:50 and cultured on type-I collagen gel-coated coverslips. After 16 h of tetracycline treatment to induce the expression of GFP-Ras$^{V12}$ in MDCK-pTR GFP-Ras$^{V12}$ cells, all cells were fixed with 4% paraformaldehyde in phosphate-buffered saline (PBS) and permeabilized in 0.5% Triton X-100 in PBS. After blocking the cells using 1% bovine serum albumin in PBS, cells were treated with Alexa Fluor 568 phalloidin (Invitrogen, #A12380). The nuclei were counterstained using 4′, 6-diamidino-2-phenylindole (DAPI), and coverslips were mounted using ProLong Diamond antifade mountant (Life Technologies). Images were obtained using an LSM710 confocal microscope (Zeiss). As shown in Fig. 2c, MDCK-pTR GFP-Ras$^{V12}$ cells were treated using 50 μM dichloroacetate or 250 ng/mL of each recombinant protein for 3 days and then cell competition assay with normal MDCK (parent) cells was performed as described above. For crizotinib treatment experiments, MDCK-pTR GFP-Ras$^{V12}$ cells were treated with 20 ng/mL recombinant HGF (Supplementary Fig. 6b) or CM derived from Ras$^{V12}$-induced senescent IMR-90 cells (Supplementary Fig. 6c) for 3 days. An hour before performing the cell competition assay, MDCK-pTR GFP-Ras$^{V12}$ cells were treated with 100 nM crizotinib, followed by a cell competition assay with normal MDCK (parent) cells as described above.

**RNA interference.** HGF knockdown was performed via the transfection of siRNAs using Lipofectamine RNAiMAX Transfection Reagent (Thermo Fisher Scientific) according to the manufacturer's instructions. ON-TARGETplus human HGF siRNA (Dharmacon, #LQ-006650-00-0002) was used at a concentration of 25 nM for 2 days. Knockdown efficiency was evaluated using real-time quantitative polymerase chain

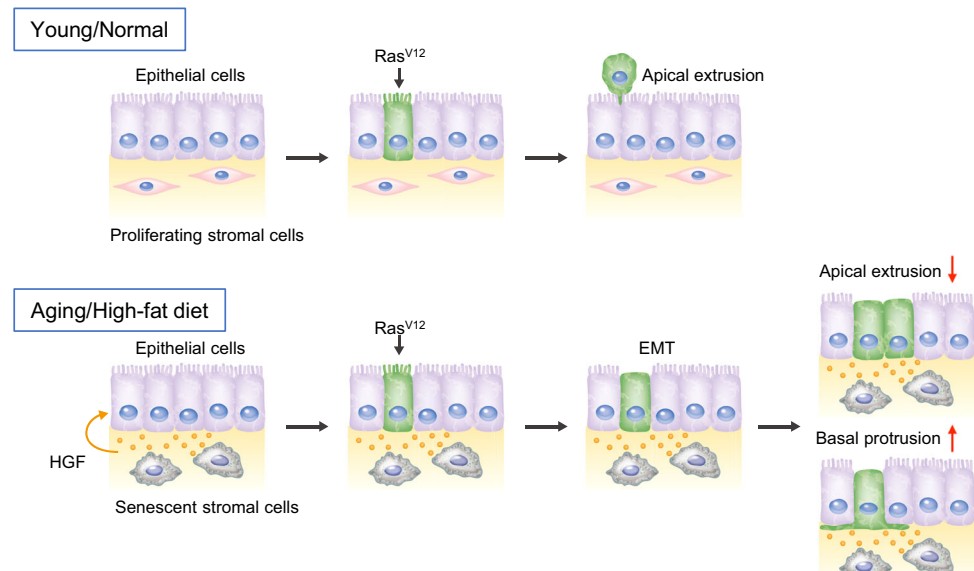

**Fig. 6 A model of cell competition in young and aged tissues.** In normal tissues, Ras[V12]-transformed cells are removed via apical extrusion from epithelial layers. However, senescent cells accumulate in various tissues with aging or consumption of a high-fat diet. Consequently, HGF secreted from senescent stromal cells suppresses apical extrusion and promotes basal protrusion of Ras[V12]-transformed cells, leading to abnormal cell proliferation in epithelial layers and infiltration to basal layers. Therefore, the decline of cell competition might be etiologically associated with the development of cancer in aged tissues.

reaction (RT-qPCR). For the preparation of CM from IMR-90 cells with HGF knockdown, proliferating IMR-90 cells were exposed to 10-Gy irradiation using a CP-160 X-ray machine. These cells were plated at a density of 2500 cells cm⁻² and were not passaged after X-ray irradiation. After 8 and 9 days of X-ray irradiation, HGF knockdown was performed (twice) as described above. Ten days after X-ray irradiation (1 day after the final knockdown of HGF), IMR-90 cells were cultured in fresh DMEM supplemented with 10% FBS and incubated for 3 days. CM was filtered through a 0.45-μm filter and then used for cell culture as HGF-depleted senescent CM.

**RT-qPCR**. Total RNA was extracted using the *mir*Vana™ miRNA Isolation Kit (Thermo Fisher Scientific). Then, the extracted RNA was subjected to reverse transcription using the PrimeScript RT Master Mix (TaKaRa). RT-qPCR was performed on a StepOne Plus PCR system (Thermo Fisher Scientific) using SYBR *Premix Ex Taq* II (Tli RNaseH Plus, TaKaRa, #RR820A). The primers used for RT-qPCR are as follows: HGF (human): 5′-GCACTGAAGATAAAAACCAA-3′ (forward) and 5′-GTTTTCTCGCTTTATCAAAA-3′ (reverse), HGF (mouse): 5′-GGCCCACTCATTTGTGAAC-3′ (forward) and 5′-CATCCACGACCAGGAAC-3′ (reverse), p21 (mouse): 5′-CCTGGTGATGTCCGACCTG-3′ (forward) and 5′-CCATGAGCGCATCGCAATC-3′ (reverse)[45], β-Actin (mouse): 5′-CGCCAC-CAGTTCGCCATGGA-3′ (forward) and 5′-ACAGCCCGGGGAGCATCGT-3′ (reverse), CDH1: 5′-CAGAAGATGACACCCGGGAC-3′ (forward) and 5′-GCCACATCATTGCGAGTCAC-3′ (reverse), ZO1: 5′-CCACACCGCTG GTTTTAAGC-3′ (forward) and 5′-TCTTCGGGTGGCTTCATCTG-3′ (reverse), Fibronectin (dog): 5′-AGCAAATGGCCAGAATCCGA-3′ (forward) and 5′-CTTGTAGTCAGTGCCGGGTT-3′ (reverse), ZEB1: 5′-GAAGGTGATCCAG-CAAATG-3′ (forward) and 5′-CTTCCGCATTTTCTTTTTGG-3′ (reverse), β-catenin (dog): 5′-TGTATGGGTAGGGTAAATCAGGAAT-3′ (forward) and 5′-TGTATGGGTAGGGTAAATCAGGAAT-3′ (reverse), and β-Actin (dog): 5′-GGCACCCAGCACAATGAAG-3′ (forward) and 5′-ACAGTGAGGCCAG-GATGGAG-3′ (reverse). p16, IL-6, IL-8, CXCL10, β-actin (human), p16, CXCL10, and GAPDH (mouse) were detected as previously described[30]. The quantity of all samples was acquired using the standard curve method and was normalized to the housekeeping gene *ACTB* and *GAPDH* according to the manufacturer's protocol.

**Western blotting**. Cell pellets were lysed in lysis buffer (0.1 M Tris-HCl (pH 7.5), 10% glycerol, and 1% sodium dodecyl sulfate (SDS)), boiled for 5 min, and then centrifuged for 10 min at 15,000 rpm. All protein concentrations were determined using BCA Protein Assay Reagent (Pierce). Each cell lysate was electrophoresed using SDS-polyacrylamide gel electrophoresis and transferred onto polyvinylidene difluoride membranes (Millipore). After blocking with 5% skim milk (Megmilk) or 5% bovine serum albumin (Sigma-Aldrich) in Tris-buffered saline with 0.1% Tween 20 (TBST), the membrane was treated with primary antibodies against Ras (1:1000, Oncogene Research Products, #OP41), lamin-B1 (1:1000, Abcam, #ab16048), GAPDH (1:10,000, Proteintech, #60004-1), and phospho-Met (Tyr1234/1235) (D26) (1:1000, Cell Signaling Technology, #3077) overnight at 4 °C in a blocking buffer. Membranes were then washed thrice in TBST and incubated

with enhanced chemiluminescence (ECL) anti-mouse IgG, horseradish peroxidase-linked whole antibody (GE Healthcare, #NA931V) or ECL anti-rabbit IgG, and horseradish peroxidase-linked whole antibody (GE Healthcare, #NA934V) for 1 h at room temperature. After washing the membrane thrice with TBST, the signal was resolved using SuperSignal West Femto Maximum Sensitivity Substrate (Thermo Fisher Scientific) and imaging was performed using a FUSION imaging system (Vilber-Lourmat).

**Enzyme-linked immunosorbent assay (ELISA) and cytokine array**. CM from proliferating (control) and X-ray- or Ras-induced senescent IMR-90 cells was used for performing ELISA and cytokine array. The ELISA and cytokine array were performed using Quantikine ELISA Human HGF Immunoassay (R&D Systems, #DHG00B) and Proteome Profiler Array Human XL Cytokine Array Kit (R&D Systems, #ARY022B), respectively, according to the manufacturer's instructions. In the cytokine array, signal intensity was detected using a FUSION imaging system (Vilber-Lourmat) and quantified using NIH Image J software. The signal intensities of six reference spots in each membrane were measured and defined as 100%.

**Immunofluorescence imaging**. Immunofluorescence analysis was conducted using antibodies against γ-H2AX (1:2000, Millipore, #05–636) and phospho-(Ser/Thr) ataxia telangiectasia mutated (ATM)/ ataxia telangiectasia and Rad3-related (ATR) substrate (1:5000, Cell Signaling Technology, #2851) as previously described[30]. Biopsies of mouse liver were fixed in 10% formalin for 24 h, progressively dehydrated through gradients of alcohol, and embedded in paraffin. Samples were sectioned using a microtome (3-μm thick) for antibody staining, deparaffinized in xylene, rehydrated, and then exposed to heat-induced antigen retrieval for 20 min in 1× Target Retrieval Solution (DAKO, #S1699). After washing with PBS, the sections were incubated in 1× Power Block (Biogenex, #HK085-5K) for 10 min at room temperature. After blocking and washing with PBS, the sections were incubated with primary antibodies overnight at 4 °C. The primary antibodies used for mouse samples were as follows: α-SMA (1:500, Sigma, #A5228), vimentin (1:250, Abcam, #ab92547), desmin (1:200, Sigma, #D1033), HGF (1:100, R&D Systems, #AF-294-NA), CXCL10 (1:100, R&D Systems, #AF-466-NA), 53BP1 (1:500, Santa Cruz, #sc-22760) and phalloidin (1:100, Thermo Fisher Scientific, #A12380). Alexa Fluor 488 goat anti-mouse (1:500, Thermo Fisher Scientific, #A12380). Alexa Fluor 594 goat anti-rabbit (1:500, Thermo Fisher Scientific, #A11012), and Alexa Fluor 594 donkey anti-goat antibodies (1:500, Thermo Fisher Scientific, #A11058) were used as secondary antibodies. SA-β-Gal activity with Ki-67 protein was detected using a Cellular Senescence Detection Kit and SPiDER-β-Gal (Dojindo, #SG03) with recombinant anti-Ki-67 antibody (SP6) (Abcam, #ab16667), respectively, according to the manufacturer's instruction.

**Microarray analysis**. Total RNA was extracted from MDCK-pTR GFP-Ras[V12] cells that were treated with CM derived from proliferating (control) and X-ray- or Ras-induced senescent IMR-90 cells (the details are provided in the "Cell Culture"

section) for 3 days using *mirVana*™ miRNA Isolation Kit (Thermo Fisher Scientific) according to the manufacturer's protocols. Microarray analysis was conducted using the Plant and Animal Gene Expression platform (Agilent). Labeled cRNA was prepared from 1 to 5 µg of total RNA using Agilent's Quick Amp Labeling Kit. Following fragmentation, 1.65-µg cRNA was hybridized using the Agilent expression microarray according to the manufacturer's protocols. Arrays were scanned using the Agilent Technologies G4900DA SG12494263. Array data export processing and analysis were conducted using Agilent Feature Extraction v11.0.1.1. GO analysis was performed using DAVID[46].

**Plasmids**. pSBbi-Hyg-EGFP-Luc (control) and pSBbi-Hyg-N-Ras$^{V12}$-IRES-EGFP-Luc plasmids were constructed as shown below: human G12V N-Ras (N-Ras$^{V12}$)[47] and IRES-EGFP with luciferase[12] sequences were cloned into a pSBbi-Hyg plasmid (Addgene, #60524). The IRES-EGFP sequence was based on a pIRES2-EGFP plasmid (Invitrogen).

**HTVi and immunohistochemistry**. Male C57BL/6 mice (CLEA Japan Inc.) of age 7–8 weeks were maintained under specific pathogen-free conditions and fed ND (CE-2 from CLEA Japan, 12 kcal% of fat) or a high-fat diet (HFD, D12492 from Research Diets, 60 kcal% of fat) for 3 months. The room was maintained at a controlled temperature (~25 °C) and humidity (~50%) and 12 h light–dark cycles. HTVi was conducted as previously described[26,42] with minor modifications. In brief, 27-µg pSBbi-Hyg-EGFP-Luc or pSBbi-Hyg-N-Ras$^{V12}$-IRES-EGFP-Luc plasmid and 3-µg pCMV(CAT)T7-SB100 plasmid (Addgene, #34879) (total 30-µg DNA per mouse) were injected into the tail vein using TransIT-EE Hydrodynamic Delivery Solution (Mirus) according to the manufacturer's protocols. Mice were orally administered with 50 mg/kg crizotinib (LKT laboratories, #C6935) at time points indicated in Fig 4a. In vivo bioluminescent imaging was conducted using the IVIS Imaging System (Perki-nElmer). The intensity of luminescence (total Flux (p/s)) after 6 days of HTVi was calculated using Living Image Software (PerkinElmer). The relative luminescence was normalized by the average luminescence intensity of each empty control-injected mouse. After 1 and 6 days of HTVi, isolated mice livers were embedded in Tissue-Tek OCT Compound (Sakura, #4583). Then, 5-µm-thick frozen sections were cryo-sectioned on a glass slide (Matsunami, #FRC-15). The sections were fixed with 4% paraformaldehyde in PBS. Immunohistochemistry was performed using the primary antibodies against NRAS (1:100, Proteintech, #10724-1-AP), GFP (1:700, Abcam, #ab13970), E-cadherin (1:700, Invitrogen, #131900). After blocking with 10% FBS in PBS, antibodies were incubated in 1% FBS overnight. The slides were then stained with secondary antibodies, such as goat anti-rabbit IgG (H + L) Cross-Adsorbed Secondary Antibody, Alexa Fluor 594 (1:500, Thermo Fisher Scientific, #A11012), and goat Anti-chicken IgY H&L (Alexa Fluor 488) (1:500, Abcam, #ab150169). Nuclei were counterstained with DAPI, and coverslips were mounted using ProLong Diamond antifade mountant (Life Technologies, #P36961). Images showing fluorescence were observed and photographed using an immunofluorescence microscope (Carl Zeiss AG). All animal procedures were conducted using protocols approved by the JFCR Animal Care and Use Committee, according to the relevant guidelines and regulations (approval number: 1804-05).

**In vivo and ex vivo cell competition model**. In Fig. 5 and Supplementary Figs. 10 and 11, HFD-feeding and immunofluorescence staining of cell competition model mice were performed as previously described[18,27]. For in vivo experiments, 6–10 weeks old Villin-Ras$^{V12}$-GFP mice were given a single intraperitoneal injection of 2 mg of tamoxifen in corn oil (Sigma, #C8267), and were then sacrificed days after Cre activation. For some experiments, the senolytic drug ARV825 (MedChemExpress, #HY-16954)[31] was intraperitoneally administered for 5 consecutive days (total of 2 weeks). For culturing intestinal organoids, isolated crypts from the mouse small intestine were entrapped in Matrigel (Corning, #356231) and plated in a non-coated 35-mm glass-bottom dish as previously described[48]. The crypts embedded in Matrigel were covered with Advanced DMEM/F12 supplemented with N2 (Gibco, #17502-048), B27 (Gibco, #17504-044), 50 ng ml$^{-1}$ EGF (Peprotech, #315-09), 100 ng ml$^{-1}$ Noggin (Peprotech, #250-38), 1.25 mM N-Acetylcystein (Sigma-Aldrich, #A7250), and R-spondin conditioned medium collected from 293T-HA-Rspol-Fc cells kindly provided by Dr. Calvin Kuo (Stanford University). After 96 h of culture, organoids were incubated with tamoxifen (Sigma, #T5648) for 24 h to induce transgenes. Subsequently, tamoxifen was washed out, and organoids were cultured for 24 h for analyses.

**RNA in situ hybridization**. In situ hybridization–immunohistochemistry code-tection was performed according to the standard workflow with some modifications. In brief, immunohistochemical detection was first performed via a standard polymer detection method using nestin polyclonal antibody (1:200, Proteintech, #19483-1-AP), and the immunohistochemical signal was detected using the DAB chromogen. After immunohistochemical detection, the samples were fixed in 10% neutral-buffered formalin for 30 min, and then in situ hybridization was performed using RNAscope Probe Mm-Hgf-1 (Advanced Cell Diagnostics, #435381) and RNAscope 2.5 HD Reagent KIT-RED.

**Statistical analysis**. Parametric statistical analyses were conducted using the unpaired two-tailed Student's *t* test, or one-way analysis of variance (ANOVA), followed by Dunnett's or Tukey's multiple comparisons posthoc test using the R software for statistical computing (64-bit version 3.6.1). Nonparametric statistical analyses were conducted using the Wilcoxon rank-sum or chi-square test using the R software for statistical computing. A *P*-value of <0.05 was considered statistically significant.

**Reporting summary**. Further information on research design is available in the Nature Research Reporting Summary linked to this article.

## Data availability

The microarray data were deposited in the DNA Data Bank of Japan with the accession number E-GEAD-448. All data are available within the Article, Supplementary Information or Source Data file. Source data are provided with this paper.

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

## Acknowledgements

We thank R.S. Nozawa for technical assistance for the confocal microscopy analysis and the members of Takahashi's laboratory for helpful discussion during the preparation of this manuscript. This work was supported in part by grants from Japan Agency of Medical Research and Development (AMED)-PRIME under grant number 19gm6110023h0001, Japan Science and Technology Agency (JST)-Moonshot R&D under grant number JPMJPS2022, Japan Society for the Promotion of Science (JSPS) under grant number 17H05628 and 19K22571, The Cell Science Research Foundation, and Astellas Foundation for Research on Metabolic Disorders.

## Author contributions

A.T. conceived and designed the experiments. N.I., K.M., T.M.L., M.C., A.H., M.N., H.K., H.Z., and S.K. performed the experiments. N.I., K.M., T.M.L., S.T., S.K., K.M., Y.F., and A.T. analyzed the data. N.I., K.M., T.M.L., and A.T. wrote the manuscript, and A.T. oversaw the projects. All the authors approved the manuscript.

## Competing interests

The authors declare no competing interests.
