## [Peer Review File · Nature Communications]

Hepatocyte growth factor derived from senescent cells attenuates cell competition-induced apical elimination of oncogenic cellsREVIEWER COMMENTS

Reviewer #1 (Remarks to the Author):

The paper by Igarashi et al, entitled, "Hepatocyte growth factor derived from senescent cells attenuates cell competition-induced apical elimination of oncogenic cells" offers important insight into how cellular senescence may impact tumour formation and progression. They find that conditioned medium from senescent fibroblast cells reduces the loss of H-Ras cells by extrusion during cell competition with wild type MDCK cells. By cytokine array analysis, they identified HGF as the SASP that was critical for reducing the apical extrusion and, thus, cell competition in Ras mutated cells, while promoting basal protrusions in these cells. An inhibitor of HGF, Crizotinib, can promote apical extrusion of H-Ras cells both in MDCKS and in mice fed a high fat diet. These findings are interesting and merit publication, following firming up some of the current data.

- The data on inducing senescence aren't very strong, as the markers that use are not that specific for senescence but can also be seen during inflammation or DNA damage. They should also include the gold standard of SA-Beta-gal with loss of Ki67 staining.
- In figure 1, the reduction of extrusion does not seem very impressive. It is not clear from the graphs what the numbers are-percentages from an experiment? Total numbers of extruding cells. I find it hard to interpret how significant this is.
- In Fig. 2, they show that HGF blocks apical extrusion and promotes basal protrusions, but it would be useful to see representative pictures of what this means in each category that they quantify. It is not very clear here or in later in vivo sections what the relevance of basal protrusions are? Do these cells invade later in Fig. 4?
- In vivo data in Fig. 4 should confirm if Crizotinib affects HGF/c-met signalling. Also, pictures in Fig. 4 would benefit from higher magnification larger presentation. It is very hard to see what they are imaging or what they are focusing on.
- Introduction and discussion are focused on cancer, but the results do not really touch on cancer, rather transformed cells. The authors should scale back their claims on cancer. Moreover, it is not clear from Fig. 4 what the relevance is to tumour formation. How relevant is HRas mutation to liver carcinomas? How relevant is senescence to its prevalence as well. The HFD is interesting from mice, but high fat diets in humans, tend to be associated with ketonic diets, and do not cause obesity as seen in mice. Is there an equivalent treatment that would drive senescence in humans and is this linked with carcinoma formation?

Reviewer #2 (Remarks to the Author):

In this exciting MS by Fujita, Takahashi and colleagues, the study the influence of senescent cells on cell competition. This is a long overdue line of research, because senescent and unfit cells, cell competition and senescence, have similarities and are involved in several fields, most importantly cancer and ageing. Here they find that one factor secreted by senescent cells, hepatocyte growth factor (HGF), inhibits the apical extrusion and promotes the basal protrusion of Ras mutated cells in the cell competition assay. This suggests that senescence could shift the role of cell competition from tumor suppression to tumor promotion. Although many questions remain open, I think the MS is outstanding and could encourage further work on the interconnections between senescence and cell competition. The results are convincing and I do not have major problems and support publication. I have a few minor comments that could be addressed by the authors in writing:

- 1- - In my opinion, the physiological role of tumor suppressive cell competition is not completely clear. For example, in flies, where it was initially discovered, *eiger* (the TNF homologue in *Drosophila*) is absolutely required to eliminate tumor cells in experimentally induced tumors. However, *eiger* mutant flies are viable, and there is no evidence that they have more tumors during a normal life, so the normal physiological role of tumor suppression cell competition is not clear in less artificial (more physiological) conditions. It would be great if the authors discuss this also regarding mammals, what is the evidence that tumor suppressive cell competition works under physiological conditions? This could impact their conclusions, because live extruded cells may metastasize rather than just be eliminated and the direction of extrusion could make a difference in the outcome. For instance, are apically or basally extruded cells more prone to die or

to survive, and if surviving, could they metastasize? In my opinion it is still an open question whether extruded cells are eliminated or metastasizing. I would love to know the opinion of the authors and learn about the evidence.

2- When citing the role of cell competition in ageing, they cite a review, which is OK, but the pioneering work that first connected ageing and cell competition should also be cited: Merino et al., *Cell*, 2015.

3- Although I have read other MS by the authors and they are nicely written, I think the English writing of this particular MS should be improved.

Reviewer #3 (Remarks to the Author):

Cell competition is a novel anti-cancer mechanism and the manuscript by Igarashi et al., provides new data to further explore this subject. The angle to research cell competition in the context of cellular senescence is interesting and relevant as senescent cells have been shown to contribute to tumorigenesis and aging. However, as much as authors provided reliable preliminary evidence on the relationship between SASP and cell competition *in vitro*, the *in vivo* part is relatively weak.

Major comments:

1. One thing which is unclear to me for the *in vitro* part is the reasoning behind the selection of the senescence inducers for specific experiments. For the experiments presented in Fig. 1, CM from OIS IMR90 and from replication-induced senescence of TIG3 cells was used. For the Fig. 2 HGF from X-ray induced IMR90 cells was quantified and the data on its concentration from that experiment was used to validate the hypothesis on the HGF-driven reduction in cell competition. To strengthen the hypothesis all the combinations should be used in both figures: at least 2 types of cells induced to senescence via three routes (OIS, DIS and replicative senescence). This experiment could show that different cell types and different senescence inducers result in different quantities of HGF and thus variable effects on apical exclusion. Overall, a correlation between HGF secretion in different cell types and types of senescence would provide an important evidence to further strengthen the hypothesis.
2. It is unclear to me what is the link between reduction in apical exclusion of epithelial cells and the EMT. Is induction of EMT the sole mechanism by which HGF inhibits cell competition? How does it work? Why is the induction of EMT specific only to the Ras-transformed cells and not the surrounding epithelium? I would suggest performing more experiments to better understand this connection. Among other experiments, the authors could test whether other interventions inducing EMT have effects on cell competition similar as HGF. Similarly, expression levels of other EMT-related genes such as vimentin, fibronectin and catenin could be tested. Cadherins and occludins are usually associated with better adherence to cell layers so a decrease in expression of these genes could indicate lower retention of cells in the epithelial layers while the opposite has been observed. In opinion of this Reviewer this part is too preliminary and requires more research to establish the relationship between the EMT, senescence and cell competition.
3. The weakest point of the manuscript in my opinion is on the impact of senescence on cell competition *in vivo*. First, the presented evidence on HFD-induced senescence in HSCs is not reliable. Only a single marker of senescence has been assessed, while there is a recommendation to look at several (>3) senescence markers, but even this marker resembles unspecific staining more than the proper expression pattern. p21 is predominantly intranuclear being relatively evenly distributed in the nucleoplasm and the "blob" shown in Supp. Fig. 5 c looks nothing like a reliable p21 staining. As a side note, the antibody used by the authors should be specific to human p21, but I found no indication it would work for murine p21. In this respect, it is recommended that authors perform experiments involving an assessment of frequency of cells bearing senescence markers including p21, p16, SA- β -gal, DNA damage and others, ideally via two methods like RT-PCR and IHC/IHF. Also, I fail to understand why HGF would be present in HSCs in form of large cytoplasmic inclusions (Supp. Fig. 5c). It might be even that the antibody (AB-294-NA) is not suitable for any form of immunofluorescence nor it is specific to mouse HGF. In my and my colleagues' experience soluble secretory factors such as HGF are very challenging to stain using antibodies. My recommendation would be to either sort senescent HSCs and perform RT-PCR to determine HGF expression or to perform RNA-ISH to quantify level of intracellular, HGF-encoding transcripts.
4. In the same part, the evidence on senescence being involved in HFD-induced reduction in cell

competition is lacking. HFD can increase levels of HGF via other means than cellular senescence and one pharmacologic intervention on inhibition of an HGF receptor is insufficient. As minimum, a set of interventions should be performed to remove senescent cells by using transgenic mouse models (e.g. p16-ATTAC, p16-3MR, p16-Rosa26 or others) and drugs targeting senescent cells (Navitoclax, Dasatinib and Quercetin or others), but ideally the experimental design would involve a selective elimination of senescent HCS cells or a targeted incapacitation of HGF production/secretion by senescent HCS. Finally, these and previous experiments should be done in other in vivo models of senescence induction such as X-ray irradiation, doxorubicin, aging or others.

5. For both models of cell competition in vivo it should be investigated whether the effects of HGF are mediated via induction of the EMT. This can be done via usage of specific and well-characterized antibodies against EMT marker proteins, RNA (or single-cell RNA) sequencing or others.

Minor comments:

1. How authors decided what quantities of peptides should be used in Fig 2c? It seems to me that unless several concentrations of each peptide were tested, the authors might have missed the window of the effective activity. 2C also seems to be missing statistics.

2. aSMA is insufficient to mark HSCs as this protein in liver marks also smooth muscle cells, among others. More stainings/assays are needed to show co-localization between senescence markers and HSC markers.

Point-by-point responses to the reviewers' comments

We would like to thank all the three reviewers for their valuable comments and constructive suggestions. We have tried to address all the issues that they have noted and believe that the current manuscript has been significantly improved.

Reviewer #1:

The paper by Igarashi et al, entitled, "Hepatocyte growth factor derived from senescent cells attenuates cell competition-induced apical elimination of oncogenic cells" offers important insight into how cellular senescence may impact tumour formation and progression. They find that conditioned medium from senescent fibroblast cells reduces the loss of H-Ras cells by extrusion during cell competition with wild type MDCK cells. By cytokine array analysis, they identified HGF as the SASP that was critical for reducing the apical extrusion and, thus, cell competition in Ras mutated cells, while promoting basal protrusions in these cells. An inhibitor of HGF, Crizotinib, can promote apical extrusion of H-Ras cells both in MDCKS and in mice fed a high fat diet. These findings are interesting and merit publication, following firming up some of the current data.

Response:

We appreciate your constructive insights.

1) The data on inducing senescence aren't very strong, as the markers that use are not that specific for senescence but can also be seen during inflammation or DNA damage. They should also include the gold standard of SA-Beta-gal with loss of Ki67 staining.

Response-1:

We agree that this is an important point. To further strengthen our data, we conducted SA- β -Gal with loss of Ki67 staining. We confirmed the induction of senescent cell cycle arrest in all experiments (see new Fig. 1c, Extended Data Fig. 1b, 2a, d, g, and 3b).

2) In figure 1, the reduction of extrusion does not seem very impressive. It is not clear from the graphs what the numbers are-percentages from an experiment? Total numbers of extruding cells. I find it hard to interpret how significant this is.

Response-2:

We apologize for having caused this confusion. We have conducted several additional experiments and statistical analyses to address this point. We have counted more than one hundred Ras^{V12}-expressing cells per experiment and decided the percentage of apical extruded, extruding, or not extruded cells. We have repeated the cell competition assay three times independently. Then we evaluated the effect of culture supernatant from senescent cells. CM derived from senescent cells increased the number of not extruded MDCK/Ras cells and significantly decreased the number of extruded MDCK/Ras cells (see the revised Fig. 1f). Additionally, we performed cell competition analysis via three routes (OIS, DIS and replicative senescence) using two types of cells (TIG-3 and IMR-90 cells) (see new Extended Data Fig. 1e, 2j, and 3d) and performed at least two biological replicates in all experiments. Consequently, we confirmed that senescent cell-derived culture supernatants suppressed the efficiency of apical extrusion of Ras^{V12}-transformed cells under all conditions.

3) In Fig. 2, they show that HGF blocks apical extrusion and promotes basal protrusions, but it would be useful to see representative pictures of what this means in each category that they quantify. It is not very clear here or in later *in vivo* sections what the relevance of basal protrusions are? Do these cells invade later in Fig. 4?

Response-3:

According to your suggestion, we showed representative pictures of what this means in each category that we quantified (see the revised Fig.2c). Since it is technically difficult to evaluate basal protrusion *in vivo*, we used intestinal organoids established from the cell competition mouse model. In this model, “basal extrusion,” in which cells completely translocated under the epithelial cell layer and invaded into the basal membrane, was observed instead of basal protrusion in MDCK cells (Kon *et al.*, Nature Cell Biol., 2017; Sasaki *et al.*, Cell Rep., 2018). Coculturing organoids with senescent fibroblasts (IMR-90) significantly inhibited apical extrusion and promoted basal extrusion of Ras^{V12}-transformed epithelial cells compared with coculturing organoids with nonsenescent fibroblasts (see the Figure below). Moreover, we examined the fate of Ras^{V12}-expressing cells *in vivo* using a cell competition mouse model. Although it was a minor population, we also observed the increase in basal extrusion of Ras^{V12}-expressing cells in the small intestine of HFD-fed mice (see new Extended Data Fig. 11). Furthermore, this phenomenon was

attenuated by the administration of ARV825, a snolytic drug that specifically eliminates senescent cells (Wakita *et al.*, Nature Commun., 2020) (see new Extended Data Fig. 11). Therefore, we consider that basal protrusion observed in the MDCK model treated with senescent CM might be relevant to the *in vivo* cell competition model fed HFD.

Figure legend:

Immunofluorescence images of intestinal organoids of villin-CreERT2, LSL-Ras^{V12}-IRES-eGFP mice after treatment with 100 nM tamoxifen coculturing with nonsenescent IMR90 cells (control) or Ras-induced senescent IMR90 cells (HRas^{V12}). “Extruding”: with their nucleus apically shifted, but still attached to the basement membrane. “Apical extruded”: completely detached from the basement membrane and translocated into the apical lumen. “Basal extrusion”: completely translocated under the epithelial cell layer and invading the basal membrane. The white arrowhead indicates basally extruded Ras^{V12}-GFP cells. Data are presented as mean ± standard error of the mean. **P* < 0.05 for extruded + extruding Ras^{V12}-GFP cells coculturing with control or senescent IMR-90 cells by unpaired two-tailed t-test; three independent experiments.

4) *In vivo* data in Fig. 4 should confirm if Crizotinib affects HGF/c-met signalling.

Response-4:

In line with your suggestion, we evaluated the effect of crizotinib treatment on HGF/c-met signaling using immunohistochemistry. As a result, we observed that the levels of phospho-c-met signals increased in the liver of HFD-fed mice than in the liver of ND-fed mice. Expectedly, crizotinib treatment caused a significant reduction in phospho-c-met signals in the liver (see new Extended Data Fig. 9c). Similarly, the downregulation of phospho-c-met signals was also observed in the small intestines of cell competition mice after crizotinib treatment (see new Extended Data Fig. 10).

Also, pictures in Fig. 4 would benefit from higher magnification larger presentation. It is very hard to see what they are imaging or what they are focusing on.

Response-5:

We thank you for this advice. Accordingly, we have provided higher-magnification images in Fig. 4C.

5) Introduction and discussion are focused on cancer, but the results do not really touch on cancer, rather transformed cells. The authors should scale back their claims on cancer. Moreover, it is not clear from Fig. 4 what the relevance is to tumour formation. How relevant is HRas mutation to liver carcinomas? How relevant is senescence to its prevalence as well. The HFD is interesting from mice, but high fat diets in humans, tend to be associated with ketonic diets, and do not cause obesity as seen in mice. Is there an equivalent treatment that would drive senescence in humans and is this linked with carcinoma formation?

Response-6:

We highly appreciate your valuable suggestions regarding our manuscript. H-Ras^{V12} mutation has been associated with liver carcinomas (Sui *et al.*, *Oncol Lett.*, 2012). In addition, other groups and we have reported that HFD induced cellular senescence in HSCs and promoted hepatocellular carcinoma (HCC) development via H-Ras^{V12} mutation in the liver of obese mice (Yoshimoto *et al.*, *Nature*, 2013; Loo *et al.*, *Cancer Discovery*, 2018; Takahashi *et al.*, *Nature Commun.*, 2018; Wakita *et al.*, *Nature Commun.*, 2020). These previous findings revealed that senescent HSCs caused cancer development in the liver of obese mice via inflammatory SASP factors. Additionally, our data suggested that senescent HSCs treated with HFD and doxorubicin (DOXO) promoted the retention of Ras^{V12}-transformed cells in the epithelial layer through the inhibition of cell competition mediated by HGF, which might be a risk factor for HCC development (See Response-5 for Reviewer #3). However, we did not evaluate cancer development via a long-term study here. Therefore, we agree with your comments and have scaled back the claims on cancer in the revised manuscript on page 12, line 252 to page 13, line 269.

We believe that it is very difficult to answer whether HFD treatment is relevant for humans, because we could not assess the effect of HFD in humans. We demonstrated that HFD induced cellular senescence in the HSCs of obese mice through deoxycholic acid (DCA), a secondary bile acid. A previous report showed

that high fat consumption resulted in higher fecal DCA concentrations in healthy male volunteers (Rafter *et al.*, Am. J. Clin. Nutr., 1987). In addition, cellular senescence and SASP were also observed in HSCs obtained from patients with HCC and nonalcoholic steatohepatitis, which is closely associated with obesity (Yoshimoto *et al.*, Nature, 2013; Loo *et al.*, Cancer Discovery, 2017). Therefore, we suggest that DCA-induced senescent HSCs may contribute to obesity-associated HCC development via SASP factors observed in senescent HSCs in humans.

Reviewer #2:

In this exciting MS by Fujita, Takahashi and colleagues, the study the influence of senescent cells on cell competition. This is a long overdue line of research, because senescent and unfit cells, cell competition and senescence, have similarities and are involved in several fields, most importantly cancer and ageing. Here they find that one factor secreted by senescent cells, hepatocyte growth factor (HGF), inhibits the apical extrusion and promotes the basal protrusion of Ras mutated cells in the cell competition assay. This suggests that senescence could shift the role of cell competition from tumor suppression to tumor promotion. Although many questions remain open, I think the MS is outstanding and could encourage further work on the interconnections between senescence and cell competition. The results are convincing and I do not have major problems and support publication. I have a few minor comments that could be addressed by the authors in writing:

Response:

We thank you for the constructive insights.

1) In my opinion, the physiological role of tumor suppressive cell competition is not completely clear. For example, in flies, where it was initially discovered, eiger (the TNF homologue in *Drosophila*) is absolutely required to eliminate tumor cells in experimentally induced tumors. However, eiger mutant flies are viable, and there is no evidence that they have more tumors during a normal life, so the normal physiological role of tumor suppression cell competition is not clear in less artificial (more physiological) conditions. It would be great if the authors discuss this also regarding mammals, what is the evidence that tumor suppressive cell competition works under physiological conditions? This could impact their conclusions, because live extruded cells may metastasize rather than just be eliminated and the direction of extrusion could make a difference in the outcome. For instance, are apically or basally extruded cells more prone to die or to survive, and if surviving, could they metastasize? In my opinion it is still an open question whether extruded cells are eliminated or metastasizing. I would love to know the opinion of the authors and learn about the evidence.

Response-1:

We thank you for this important comment. Maybe the text about physiological role of tumor-suppressive cell competition was not clear enough. However, from the analysis using a transgenic mouse model, if the elimination of oncogenic mutant cells

by cell competition was inhibited in some pathological conditions such as obesity or inflammation, the oncogenic (cancer-prone) mutant cells were observed to accumulate in the mouse body (Kon *et al.*, Nature Cell Biol., 2017; Sasaki *et al.*, Cell Reports, 2018; Sato *et al.*, Commun. Biol., 2020). Recently it has been reported that most premalignant mutant cells were eliminated via cell competition with mutant clones in the surrounding normal epithelium in a mouse model of oesophageal carcinogenesis (Colom *et al.*, Nature, 2021). Since it has been known that some lifestyle-related risk factors and pathological conditions, such as obesity or inflammation, increase the incidence of esophageal cancers, we think that cell competition might play an important physiological role in tumor suppression in mammals. According to your suggestion, we have discussed this point in the revised manuscript on page 12, line 252 to page 13, line 269.

Insufficient Dpp uptake by minute mutant cells compared to that by wild-type cells induces apoptosis, resulting in tissue elimination (Moreno *et al.*, Nature, 2002). Additionally, in mammalian cells, we reported that apically extruded Ras^{V12}-mutated cells underwent apoptosis in the lung, pancreas, and small intestine (Kon *et al.*, Nature Cell Biol., 2017). On the other hand, cell death was not observed in basally extruded cells. However, their long-term fate is still unclear and might be associated with the metastasis of Ras^{V12}-transformed cells. Therefore, eliminating cancer-prone cells by apical extrusion from the monolayer of normal epithelial cells is important for tumor-suppressive cell competition. This study demonstrated that a SASP factor, HGF, secreted by senescent stromal cells suppressed apical extrusion and promoted basal protrusion (or basal extrusion *in vivo*). The accumulation of senescent cells in the body with age might increase the risk of transformed cell expansion during early carcinogenesis.

2) When citing the role of cell competition in ageing, they cite a review, which is OK, but the pioneering work that first connected ageing and cell competition should also be cited: Merino *et al.*, Cell, 2015.

Response-2:

Thank you for your useful remarks. In accordance with your suggestion, we have added this report (Merino *et al.*, Cell, 2015) to the revised text on page 4, line 52.

3) Although I have read other MS by the authors and they are nicely written, I think the

english writing of this particular MS should be improved.

Response-3:

Thank you for the constructive comment. A native English speaker has carefully proofread our revised manuscript to improve the English.

Reviewer #3:

Cell competition is a novel anti-cancer mechanism and the manuscript by Igarashi et al., provides new data to further explore this subject. The angle to research cell competition in the context of cellular senescence is interesting and relevant as senescent cells have been shown to contribute to tumorigenesis and aging. However, as much as authors provided reliable preliminary evidence on the relationship between SASP and cell competition in vitro, the in vivo part is relatively weak.

Response:

We thank you for your constructive insights.

Major comments:

1) One thing which is unclear to me for the in vitro part is the reasoning behind the selection of the senescence inducers for specific experiments. For the experiments presented in Fig. 1, CM from OIS IMR90 and from replication-induced senescence of TIG3 cells was used. For the Fig. 2 HGF from X-ray induced IMR90 cells was quantified and the data on its concentration from that experiment was used to validate the hypothesis on the HGF-driven reduction in cell competition. To strengthen the hypothesis all the combinations should be used in both figures: at least 2 types of cells induced to senescence via three routes (OIS, DIS and replicative senescence). This experiment could show that different cell types and different senescence inducers result in different quantities of HGF and thus variable effects on apical exclusion. Overall, a correlation between HGF secretion in different cell types and types of senescence would provide an important evidence to further strengthen the hypothesis.

Response-1:

We are grateful for your valuable suggestions regarding our manuscript. In accordance with your suggestion, we induced cellular senescence via three routes (OIS, DIS, and replicative senescence) using two types of cells (TIG-3 and IMR-90 cells). Then, we confirmed the induction of senescent cell cycle arrest via p16 expression and SA- β -Gal with loss of Ki67 staining (see new Fig. 1c, Extended Fig. 1b, 2a, d, g and 3b), activation of DNA damage signaling by immunofluorescence, and expression of SASP genes via RT-qPCR. In addition, we conducted ELISA to measure the concentration of HGF in CM under each condition. We found that HGF secretion was significantly promoted in all senescent cells even through cellular senescence was induced by all routes in both cell lines (see new Fig. 2d, Extended Fig. 1f, and 2k). Subsequently, we performed cell competition analysis using CM

derived from the senescent cells described above (see new Fig. 1f, Extended Fig. 1e, 2j, and 3d). Consistent with our finding, we confirmed that senescent cell-derived CM suppressed the frequency of apical extrusion of Ras^{V12}-transformed cells under all conditions.

2) It is unclear to me what is the link between reduction in apical exclusion of epithelial cells and the EMT. Is induction of EMT the sole mechanism by which HGF inhibits cell competition? How does it work? Why is the induction of EMT specific only to the Ras-transformed cells and not the surrounding epithelium? I would suggest performing more experiments to better understand this connection. Among other experiments, the authors could test whether other interventions inducing EMT have effects on cell competition similar as HGF. Similarly, expression levels of other EMT-related genes such as vimentin, fibronectin and catenin could be tested. Cadherins and occludens are usually associated with better adherence to cell layers so a decrease in expression of these genes could indicate lower retention of cells in the epithelial layers while the opposite has been observed. In opinion of this Reviewer this part is too preliminary and requires more research to establish the relationship between the EMT, senescence and cell competition.

Response-2:

According to your suggestion, we tested the effect of TGF- β 1 as another intervention inducing EMT on cell competition analysis (Han *et al.*, *J. Clin. Invest.*, 2005; Lamouille *et al.*, *J. Cell Biol.*, 2007; Xu *et al.*, *Cell Res.*, 2009). We observed that TGF- β 1 also induced EMT and efficiently suppressed cell competition, similar to HGF (see new Extended Data Fig. 5). Therefore, we comprehended that the induction of EMT by HGF or TGF- β 1 is important for the inhibition of cell competition, at least under our experimental conditions. Additionally, RT-qPCR analysis of fibronectin and other genes showed that the expression of EMT-related genes was also altered in Ras^{V12}-expressing MDCK cells treated with CM derived from senescent cells or HGF (see new Fig. 3c, e, and the Figure below). Moreover, HFD decreased the expression levels of E-cadherin and increased the expression levels of vimentin (see new Extended Fig. 9a and b). These data indicated that the secretion of HGF from senescent cells induced EMT in Ras^{V12}-transformed cells, leading to the inhibition of cell competition.

Figure legend:

RT-qPCR analysis of EMT markers in MDCK-pTR GFP-Ras^{V12} cells with or without 20 ng/ml of HGF. * $P < 0.05$, ** $P < 0.01$ or *** $P < 0.001$ by unpaired t -test.

Oncogenic Ras signaling regulates the expression of some EMT-associated genes through the activation of some transcription factors (Edme *et al.*, J. Cell Sci., 2002; Singh *et al.*, Cancer Cell, 2009; Shao *et al.*, Cell, 2014; Yoh *et al.*, PNAS, 2016). Moreover, it has been previously reported that the alteration of chromatin accessibility via Ras signaling and additional stimulation via TGF- β 1 or HGF acts as an important mechanism for transcriptional regulation during EMT in MDCK cells (Grünert *et al.*, Nature Rev. Mol. Cell Biol., 2003; Arase *et al.*, Sci. Rep., 2017). Moreover, we examined the expression levels of EMT markers via activation of Ras signaling and HGF treatment and found that both signaling efficiently upregulated the expression of ZEB1, a mesenchymal marker, and downregulated the expression of ZO1, an epithelial marker (see the Figure below). Based on these observations, we considered it likely that the induction of EMT via HGF or TGF- β 1 was specific to Ras^{V12}-mutated cells.

Figure legend:

RT-qPCR analysis in MDCK (parent) and MDCK-pTR GFP-Ras^{V12} cells treated with or without HGF and tetracycline treatment. * $P < 0.05$, *** $P < 0.001$, or N.S. (not significant). one-way ANOVA, followed by Tukey's multiple comparisons post hoc test.

As you indicated, the reduction of E-cadherins and occludens results in lower retention of cells in the epithelial layers in a cell-autonomous manner. However, cell competition is a non-cell-autonomous phenomenon, which does not happen

without interactions with surrounding normal epithelial cells. To establish the proper apical extrusion of Ras^{V12}-mutated cells, many factors, such as PDK4, and Rab5, or other factors are essential to eliminate Ras^{V12}-mutated cells from the normal epithelial cells (Wagstaff *et al.*, Nature Commun., 2016; Kon *et al.*, Nature Cell Biol., 2017; Saitoh *et al.*, PNAS, 2017; Matamoro-Vidal *et al.*, Current Biol., 2019). In addition, junctional changes are important for apical extrusion of loser cells (Ohsawa *et al.*, Dev. Cell, 2018); likely, the deficiency of cell polarity and junctional remodeling caused by EMT in Ras^{V12}-mutated cells may lead to failure of cell competition, and it will require further investigations. We have discussed this point in the revised manuscript on page 11, lines 217–225.

3) The weakest point of the manuscript in my opinion is on the impact of senescence on cell competition in vivo. First, the presented evidence on HFD-induced senescence in HSCs is not reliable. Only a single markers of senescence has been assessed, while there is a recommendation to look at several (>3) senescence markers, but even this marker resembles unspecific staining more than the proper expression pattern. p21 is predominantly intranuclear being relatively evenly distributed in the nucleoplasm and the “blob” shown in Supp. Fig. 5 c looks nothing like a reliable p21 staining. As a side note, the antibody used by the authors should be specific to human p21, but I found no indication it would work for murine p21. In this respect, it is recommended that authors perform experiments involving an assessment of frequency of cells bearing senescence markers including p21, p16, SA-β-gal, DNA damage and others, ideally via two methods like RT-PCR and IHF/IHC.

Response-3:

According to your helpful suggestion, we performed experiments involving an assessment of the frequency of cells bearing several senescence markers (>3). We have checked the expression of murine p21 via immunohistochemistry and RT-qPCR (see new Extended Data Fig. 8b and c). Some previous reports also used the same antibody (ab109520) to detect murine p21 via immunohistochemistry and western blotting (Lv *et al.*, Nat. Commun., 2017; Hu *et al.*, Aging Cell, 2020; Hu *et al.*, J. Cardiovasc. Transl. Res., 2022). In addition, it is well known that there is no useful antibody against murine p16 for performing immunohistochemistry without the M156 clone, which is out of stock now. Therefore, we confirmed the expression of p16 via qPCR (see new Extended Data Fig. 8c). Moreover, we detected DNA

damage signaling (53BP1) and SA- β -gal activity (see new Extended Data Fig. 8b), which are common markers for senescent cells. In addition, we confirmed SASP factor expression via immunohistochemistry and RT-qPCR (CXCL10) (see new Extended Data Fig. 8b and c). Importantly, the expression levels of HGF were significantly upregulated in the HSCs and liver tissues of HFD-fed mice (see new Extended Data Fig. 8). Therefore, we concluded that HFD induced senescence in HSCs in the liver of obese mice.

Also, I fail to understand why HGF would be present in HSCs in form of large cytoplasmic inclusions (Supp. Fig. 5c). It might be even that the antibody (AB-294-NA) is not suitable for any form of immunofluorescence nor it is specific to mouse HGF. In my and my colleagues' experience soluble secretory factors such as HGF are very challenging to stain using antibodies. My recommendation would be to either sort senescent HSCs and perform RT-PCR to determine HGF expression or to perform RNA-ISH to quantify level of intracellular, HGF-encoding transcripts.

Response-4:

We deeply apologize for this mistake. Although we have used the antibody AF-294-NA for murine HGF detection in immunofluorescence analysis, we had described the wrong production number (ab294na). We have corrected the number of antibodies in the revised manuscript. According to the previous reports, AF-294-NA can be used to detect mouse HGF via immunofluorescence analysis (Giacobini *et al.*, *The J. Neuro Sci.*, 2007; Suga *et al.*, *Stem Cells*, 2009; Zhou *et al.*, *PLoS One*, 2014). Additionally, other groups and we could detect soluble secretory factors in murine stellate cells in the liver of ND/HFD-fed mice (Yoshimoto *et al.*, *Nature*, 2013; Loo *et al.*, *Cancer Discov.*, 2018; Takahashi *et al.*, *Nat Commun*, 2018). Therefore, we employed the HGF-specific antibody to detect its expression in the mouse livers (see new Extended Data Fig. 8b). As per your suggestion, we have also conducted an RNA *in situ* hybridization analysis. The results showed that the signals of HGF mRNA were frequently detected in HSCs obtained from HFD-fed mice compared with those obtained from ND-fed mice (see new Extended Data Fig. 8a). Altogether, we concluded that the expression levels of HGF were significantly elevated in HSCs obtained from HFD-fed mice compared with those obtained from ND-fed mice.

4) In the same part, the evidence on senescence being involved in HFD-induced reduction

in cell competition is lacking. HFD can increase levels of HGF via other means than cellular senescence and one pharmacologic intervention on inhibition of an HGF receptor is insufficient. As minimum, a set of interventions should be performed to remove senescent cells by using transgenic mouse models (e.g. p16-ATTAC, p16-3MR, p16-Rosa26 or others) and drugs targeting senescent cells (Navitoclax, Dasatinib and Quercetin or others), but ideally the experimental design would involve a selective elimination of senescent HCS cells or a targeted incapacitation of HGF production/secretion by senescent HCS. Finally, these and previous experiments should be done in other *in vivo* models of senescence induction such as X-ray irradiation, doxorubicin, aging or others.

Response-5:

Since we do not have transgenic mouse models (p16-ATTAC, p16-3MR, p16-Rosa26 or others), and it takes a very long time for the introduction of new transgenic mouse models in our institute due to the COVID-19 pandemic, we were unable to experiment using those mice within the revision period. Instead, per your suggestion, we used a senolytic drug ARV825, a BET family protein degrader, to remove senescent cells *in vivo*. Previously, we reported that ARV825 treatment in HFD-fed mice caused the selective elimination of senescent HSCs from the liver with HFD-fed mice and inhibited HCC development (Wakita *et al.*, Nat Commun, 2020). In addition, we also reported that treatment of senolytic drug reduced the number of senescent fibroblasts and inhibited colorectal tumors development in the mouse model harboring a mutation of the *Apc* gene (Okumura *et al.*, Nature Commun., 2021). As expected, we have observed that the treatment of ARV825 improved the efficiency of cell competition in the liver with HFD-fed mice (see the Figure below) and in the small intestine of the cell competition mouse model (see new Extended Data Fig. 11).

Figure legend:

C57BL/6 mice were fed ND or HFD for 3 months and intraperitoneally administered 5 mg/kg of ARV825

five times per week (2 weeks). Some of the ND- and HFD-fed mice were subjected to HTVi with plasmid-encoding GFP-N-Ras^{V12}-IRES-luciferase [N-Ras^{V12}, n = 4 (ND); N-Ras^{V12}, n = 5 (HFD); N-Ras^{V12} + ARV825, n = 5 (ND); N-Ras^{V12} + ARV825, n = 7 (HFD)]. (a) The mice were euthanized and subjected to *in vivo* bioluminescent imaging to confirm the GFP-N-Ras^{V12}-IRES-luciferase expression at days 1 and 6. The values of relative luminescence (intensity of NRas^{V12}) in each group. (b) Serial sections of the liver biopsy samples were subjected to immunohistochemistry for senescence marker (p21Waf1/Cip1) and stellate cell marker (desmin). The histogram indicates the percentages of p21-positive HSCs. ****P* < 0.001, **P* < 0.05 or not significant (N.S.) by one-way ANOVA, followed by Tukey's multiple comparisons post hoc test.

Furthermore, we conducted another *in vivo* senescence model using a DNA damaging agent, doxorubicin (DOXO) (Demaria *et al.*, *Cancer Discovery*, 2017). We confirmed that DOXO treatment certainly resulted in the accumulation of senescent HSCs and inhibited cell competition, leading to an increase in Ras^{V12}-transformed cells in the liver after 6 days. Moreover, the administration of crizotinib significantly improved the efficiency of cell competition in the liver of DOXO-treated mice (see the Figure below). These data suggested that HGF secretion by senescent HSCs, which was induced by HFD and DOXO treatment, might suppress cell competition *in vivo*.

Figure legend:

C57BL/6 mice were administered 10 mg/kg doxorubicin (DOXO) intraperitoneally for 10 days. Some of the mice were orally administered 50 mg/kg of crizotinib thrice and subjected to HTVi with plasmid-encoding GFP-N-Ras^{V12}-IRES-luciferase [n = 6 (no treatment); n = 5 (DOXO); n = 7 (Crizotinib); n = 6 (DOXO + Crizotinib)]. (c) The mice were euthanized and subjected to *in vivo* bioluminescent imaging to confirm GFP-N-Ras^{V12}-IRES-luciferase expression at days 1 and 6. The values of relative luminescence (intensity of NRas^{V12}) in each group. (d) Serial sections of the liver biopsy samples were subjected to immunohistochemistry for senescence marker (p21^{Waf1/Cip1}) and stellate cell marker (desmin). The

histogram indicates the percentages of p21-positive HSCs. *** $P < 0.001$, * $P < 0.05$ or not significant (N.S.) by one-way ANOVA, followed by Tukey's multiple comparisons posthoc test.

5) For both models of cell competition *in vivo* it should be investigated whether the effects of HGF are mediated via induction of the EMT. This can be done via usage of specific and well-characterized antibodies against EMT marker proteins, RNA (or single-cell RNA) sequencing or others.

Response-6:

We agreed with you and have investigated the effects of HGF on the induction of EMT using antibodies against EMT marker proteins. E-cadherin decreased in the liver of HFD-fed mice compared with the liver of ND-fed mice. However, crizotinib maintained the levels of E-cadherin in the liver of HFD-fed mice and ND-fed mice (see new Extended Data Fig. 9a). Vimentin increased in the liver of HFD-fed mice versus that of ND-fed mice, and crizotinib inhibited the increase of vimentin in the HFD-fed mice liver (see new Extended Data Fig. 9b). Additionally, DOXO treatment also decreased the expression of epithelial marker, E-cadherin (a), and increased the expression level of mesenchymal marker, vimentin (b), in the liver of an *in vivo* DOXO senescence model (see the Figure below). We confirmed that inhibition of HGF signaling by crizotinib attenuated the expression of EMT markers (see the Figure below). We believe that these results support our conclusion.

Figure legend:

(a, b) C57BL/6 mice were intraperitoneally administered with 10 mg/kg of doxorubicin (DOXO) for 10 days. Some of the mice were orally administered with 50 mg/kg of Crizotinib thrice and subjected to HTVi with plasmid-encoding GFP-N-Ras^{V12}-IRES-luciferase, and immunofluorescence. E-cadherin (red) (a), vimentin (red) (b) and NRas^{V12} signals (green) were detected in the liver sections, and DNA was stained by DAPI (blue). Scale bar: 10 μ m.

Minor comments:

1. How authors decided what quantities of peptides should be used in Fig 2c? It seems to me that unless several concentrations of each peptide were tested, the authors might have missed the window of the effective activity. 2C also seems to be missing statistics.

Response-7:

We realized that your suggestion is appropriate. However, it was difficult to determine the appropriate concentration of each cytokine, because effective doses, such as ED50, varied widely among cell types and cytokines. According to previous reports in which HGF and other cytokines were used at 250 ng/ml concentrations for treating MDCK or other cell lines (Marra *et al.*, *Hepatology*, 1999; Hammond *et al.*, *Oncogene*, 2001; Howard *et al.*, *PLoS One*, 2011; Miya *et al.*, *Am. J. Physiol. Renal. Physiol.*, 2011; Khazali *et al.*, *Br. J. Cancer*, 2017), we standardized the concentration to 250 ng/ml for primary screening in Fig. 2c. Since HGF showed the strongest effect on cell competition, we focused on HGF. We believe that we should not exclude the possibility that other cytokines also affect cell competition in different experimental conditions. However, to further confirm the effect of HGF at a physiological level, we measured the concentration of HGF secreted from senescent cells, and it could significantly affect the cell-competitive phenomena as shown in Fig. 2d, e. Moreover, the depletion of HGF suppressed the inhibition effect of apical extrusion by senescent CM in Fig. 2g. Based on these observations, we believe that our first screening was appropriate because we could detect HGF, which can regulate cell competition.

2. aSMA is insufficient to mark HSCs as this protein in liver marks also smooth muscle cells, among others. More stainings/assays are needed to show co-localization between senescence markers and HSC markers.

Response-8:

We thank you for the suggestion. We repeated the experiment using another HSC marker, desmin. We showed co-localization between p21, a senescent marker, and desmin and α SMA via immunofluorescence staining of liver specimens (see new Extended Data Fig. 8b).

REVIEWERS' COMMENTS

Reviewer #1 (Remarks to the Author):

I have seen the revised paper and responses to my comments and feel that they have nicely addressed all the points I raised. I will say that the organoid results they included in the response are pretty amazing and they should consider including them in the paper, as I think that they are far more compelling than what you see in the monolayer on glass. Overall, it's an interesting paper that could explain why ageing and senescence can impact cancer initiation and progression and favor its publication.

Reviewer #2 (Remarks to the Author):

The authors have answered my comments satisfactorily.

Reviewer #3 (Remarks to the Author):

The manuscript by Igarashi et al has been significantly improved since the last submission. It is a very interesting piece of scientific literature and I believe will be useful for the scientific community and further progress in the fields of cellular senescence and cancer alike. There are still some minor problems with the dataset quality/presentation (see below).

Minor comments:

1. Extended figure 8b shows p21 staining and as mentioned in my previous comments I fail to understand why p21 protein would accumulate in some kind of peri-nuclear compartment as presented on the figure. To my knowledge, the last decades of research established p21 protein to be present in the nucleoplasm, sometimes present in cytoplasm, though evenly distributed. This data could be re-evaluated using different antibodies against p21 or RNA-ISH probes against the corresponding transcript. Alternatively, and as author did a very good job at providing numerous other senescence markers these data pieces and their quantifications could be removed without weakening the conclusions too much. Also, in this figure SA- β -gal is marked as green while it is blue in the provided images.
2. There seems to be some kind of a mix-up about the Extended figure 11. This is an important piece of data showing that elimination of senescent cells can facilitate removal of pre-cancerous cells. In my opinion all the related data could be added to the main figures instead of being an extended one (especially as there are currently only 4 figures, Nat Comm allows for many more than that). However, authors state that "ARV825 treatment did not significantly affect the ratio of apically extruded RasV12-expressing cells in the small intestine of ND-fed control mice (Extended Data Fig. 11). In contrast, ARV825 treatment notably elevated the frequency of apical extrusion in HFD-fed obese mice (Extended Data Fig. 11)". The figures I received show images for intestine (b), a quantification of apical exclusion presumably for liver (c). What is missing are images for liver and quantification for intestine. Also, figure legend states that scale bars are 20 μ m, while the images show 100 and 50 μ m. This should be corrected and the whole manuscript should be checked for any missing pieces of data or mis-alignments between figures'/text's description and figures' content.
3. Especially for the in vivo/in situ part, the quality of images is rather low. Fig. 4C could be improved for replacing images for some of higher resolution and a set of magnifying panels could be added to the figure. Also, Fig. 1e and 2c would benefit from being of higher resolution (less pixelated).

Point-by-point responses to the reviewers' comments

We would like to thank all the three reviewers for their valuable comments and constructive suggestions. We have tried to address all the issues that they have noted.

Reviewer #1:

I have seen the revised paper and responses to my comments and feel that they have nicely addressed all the points I raised. I will say that the organoid results they included in the response are pretty amazing and they should consider including them in the paper, as I think that they are far more compelling than what you see in the monolayer on glass. Overall, it's an interesting paper that could explain why ageing and senescence can impact cancer initiation and progression and favor its publication.

Response:

Thank you for the constructive comment. In line with the reviewer's suggestion, we added the organoid results to the new Extended Data Fig. 11.

Reviewer #2:

The authors have answered my comments satisfactorily.

Response:

Thank you very much.

Reviewer #3:

The manuscript by Igarashi et al has been significantly improved since the last submission. It is a very interesting piece of scientific literature and I believe will be useful for the scientific community and further progress in the fields of cellular senescence and cancer alike.

There are still some minor problems with the dataset quality/presentation (see below).

Response:

We thank you for your constructive insights.

Minor comments:

1. Extended figure 8b shows p21 staining and as mentioned in my previous comments I fail to understand why p21 protein would accumulate in some kind of peri-nuclear compartment as presented on the figure. To my knowledge, the last decades of research established p21 protein to be present in the nucleoplasm, sometimes present in cytoplasm, though evenly distributed. This data could be re-evaluated using different antibodies against p21 or RNA-ISH probes against the corresponding transcript. Alternatively, and as author did a very good job at providing numerous other senescence markers these data pieces and their quantifications could be removed without weakening the conclusions too much. Also, in this figure SA- β -gal is marked as green while it is blue in the provided images.

Response-1:

We thank you for this comment. In accordance with your suggestion, we removed pictures and quantifications of p21 from Extended Fig. 8b. Additionally, we correct the description of SA- β -gal in blue.

2) There seems to be some kind of a mix-up about the Extended figure 11. This is an important piece of data showing that elimination of senescent cells can facilitate removal of pre-cancerous cells. In my opinion all the related data could be added to the main figures instead of being an extended one (especially as there are currently only 4 figures, Nat Comm allows for many more than that). However, authors state that “ARV825 treatment did not significantly affect the ratio of apically extruded RasV12-expressing cells in the small intestine of ND-fed control mice (Extended Data Fig. 11). In contrast, ARV825 treatment notably elevated the frequency of apical extrusion in HFD-fed obese mice (Extended Data Fig. 11).”. The figures I received show images for intestine (b), a quantification of apical exclusion presumably for liver (c). What is missing are images for liver and quantification for intestine. Also, figure legend states that scale bars are 20um, while the images show 100 and 50um. This should be corrected and the whole manuscript should be checked for any missing pieces of data or mis-alignments between figures'/text's description and figures' content.

Response-2:

We deeply apologize for causing this confusion. We have checked all data and corrected the scale bars. As suggested by the reviewer, we added the ARV825 treatment data to the new Fig. 5 and carefully revised sentences in the manuscript on page 10, line 191 to line 207.

3) Especially for the in vivo/in situ part, the quality of images is rather low. Fig. 4C could be improved for replacing images for some of higher resolution and a set of magnifying panels could be added to the figure. Also, Fig. 1e and 2c would benefit from being of higher resolution (less pixelated).

Response-3:

We thank you for this advice. In line with the reviewer's suggestion, we have replaced images for higher-magnification and higher-resolution in Fig. 4c. We have also provided higher-resolution images in Fig. 1e and 2c.